# Genomic exploration of the journey of *Plasmodium vivax* in Latin America

**Margaux J. M. Lefebvre**[1]*, **Fanny Degrugillier**[1], **Céline Arnathau**[1], **Gustavo A. Fontecha**[2], **Oscar Noya**[3,4], **Sandrine Houzé**[5,6], **Carlo Severini**[7], **Bruno Pradines**[8,9,10,11], **Antoine Berry**[12,13], **Jean-François Trape**[1], **Fabian E. Sáenz**[14], **Franck Prugnolle**[15,16‡]*, **Michael C. Fontaine**[1,17‡]*, **Virginie Rougeron**[15,16‡]*

1 MiVEGEC, Univ. Montpellier, CNRS, IRD, Montpellier, France, 2 Instituto de Investigaciones en Microbiología, Facultad de Ciencias, Universidad Nacional Autónoma de Honduras, Tegucigalpa, Honduras, 3 Infectious Diseases Section, "Dr. Felix Pifano" Tropical Medicine Institute, Central University of Venezuela, Caracas, Venezuela, 4 Centro Para Estudios Sobre Malaria, "Dr. Arnoldo Gabaldón" High Studies Institute, Caracas, Venezuela, 5 Université de Paris, MERIT, IRD, Paris, France, 6 AP-HP, Centre National de Référence sur le paludisme, hôpital Bichat-Claude-Bernard, Paris, France, 7 Department of Infectious Diseases, Istituto Superiore di Sanità, Rome, Italy, 8 Unité parasitologie et entomologie, Institut de Médecine Tropicale du Service de Santé des Armées, Marseille, France, 9 Aix Marseille Univ, SSA, AP-HM, RITMES, Marseille, France, 10 IHU Méditerranée Infection, Marseille, France, 11 Centre national de référence du paludisme, Marseille, France, 12 Institut Toulousain des Maladies Infectieuses et Inflammatoires (Infinity), Université de Toulouse, CNRS UMR5051, INSERM UMR 1291, UPS, Toulouse, France, 13 Département de Parasitologie et Mycologie, CHU Toulouse, Toulouse, France, 14 Centro de Investigación para la Salud en América Latina, Facultad de Ciencias Exactas y Naturales, Pontificia Universidad Católica del Ecuador, Quito, Ecuador, 15 REHABS, International Research Laboratory, CNRS-NMU-UCBL, George Campus, Nelson Mandela University, George, South Africa, 16 Sustainability Research Unit, George Campus, Nelson Mandela University, George, South Africa, 17 Groningen Institute for Evolutionary Life Sciences (GELIFES), University of Groningen, Groningen, The Netherlands

‡ Co-supervised the work.
* margaux.lefebvre@ird.fr, margaux.jm.lefebvre@gmail.com (MJML); franck.prugnolle@cnrs.fr (FP); michael.fontaine@ird.fr (MCF); rougeron.virginie@gmail.com, virginie.rougeron@cnrs.fr (VR)

**Data Availability Statement:** The raw genomic data used in this study are available on NCBI (bioproject PRJNA1152751) and NCBI SSR-ID, bioproject, biosample, and sources are indicated in

## Abstract

*Plasmodium vivax* is the predominant malaria parasite in Latin America. Its colonization history in the region is rich and complex, and is still highly debated, especially about its origin (s). Our study employed cutting-edge population genomic techniques to analyze whole genome variation from 620 *P. vivax* isolates, including 107 newly sequenced samples from West Africa, Middle East, and Latin America. This sampling represents nearly all potential source populations worldwide currently available. Analyses of the genetic structure, diversity, ancestry, coalescent-based inferences, including demographic scenario testing using Approximate Bayesian Computation, have revealed a more complex evolutionary history than previously envisioned. Indeed, our analyses suggest that the current American *P. vivax* populations predominantly stemmed from a now-extinct European lineage, with the potential contribution also from unsampled populations, most likely of West African origin. We also found evidence that *P. vivax* arrived in Latin America in multiple waves, initially during early European contact and later through post-colonial human migration waves in the late 19th-century. This study provides a fresh perspective on *P. vivax*'s intricate evolutionary journey and brings insights into the possible contribution of West African *P. vivax* populations to the colonization history of Latin America.

S1 Table. The scripts are available in this github repository, as well as the genetic map used for Relate and the header file of the DIYABC-RF analysis with the scenarios: https://github.com/MargauxLefebvre/LatinAmerica_vivax.

**Funding:** This work was supported by the French National Research Agency (anr.fr): JCJC GENAD ANR-20-CE35-0003 to V.R.; MICETRAL ANR-19-CE350010 to F.P.). It was also supported by the French Institute for Public Health Surveillance (santepubliquefrance.fr): grant No CNR Paludisme to B. P. The collection of samples in Ecuador was funded by Pontificia Universidad Católica del Ecuador (www.puce.edu.ec) : grants M131416 and N131416 to F.E.S. The funders had no role in study design, data collection and analysis, decision to publish, or preparation of the manuscript.

**Competing interests:** The authors have declared that no competing interests exist.

## Author summary

Our study investigates the origins and spread of *Plasmodium vivax* in Latin America, the primary malaria parasite in this region. By analyzing genetic data from 620 *P. vivax* samples worldwide, including new samples from West Africa, the Middle East, and Latin America, we were able to reconstruct the evolutionary history of the parasite in this region. Our findings indicate that the current *P. vivax* strains in Latin America mainly descended from a now-extinct European lineage. Additionally, we found evidence suggesting that several migration waves following human migrations waves may have contributed to the parasite diversity, with possible contributions from West African populations during the transatlantic slave trade (16th to 19th century) and further input from Europe during post-colonial human migrations in the late 19th century. This study offers a fresh perspective on how historical human migrations have shaped the genetic landscape of *P. vivax* in Latin America. These insights are valuable for understanding the parasite evolution and may help inform strategies for malaria control and prevention in the future.

## Introduction

*Plasmodium vivax* is the dominant human malaria parasite in most inter-tropical countries outside sub-Saharan Africa, mainly Asia, Middle East, Latin America, and marginally certain regions of Africa [1]. One of the main reasons for its successful geographic expansion is its ability to form dormant stages in human liver cells [2]. This allows *P. vivax* to maintain itself in temperate climates, hiding in a dormant stage during the cold months when the vectors (*Anopheles* mosquitoes) are in diapause, and creating a persistent presence of parasite reservoirs [3]. In Latin America, *P. vivax* is well established from Mexico to Brazil, where it causes 70% of malaria cases [4]. Venezuela, Brazil, and Colombia are particularly affected (79% of human malaria cases in the continent) [4]. Although *P. vivax* is well established in South and Central America and is a major public health issue, the debate is still open on how, when, and from where it colonized this continent. This knowledge is necessary to develop efficient region-specific strategies for malaria control and elimination. Additionally, insights into its historical movements can help anticipate its potential spread to new areas due to factors like migration or climate change. This knowledge may also be helpful to anticipate outbreaks in regions where malaria is not yet prevalent.

In recent decades, various methodologies have been employed to trace *P. vivax* history in Latin America with different results. The study of Gerszten *et al.* [5] reported the first detection of *P. vivax* antibodies in the liver and spleen of South American mummies, dating back 3,000 to 600 years before present (ybp), through visualization of species-specific antigens by immunohistochemistry [5]. They suggested that the parasite was already present in Latin America before the arrival of European settlers and the transatlantic slave trade (spanning from 1500 to 1830). However, this result remains debatable. Indeed, antibody cross-reactivity with *Plasmodium* species has been reported [6–8], but this was not tested by the authors. Although molecular techniques are essential to validate this hypothesis [9], these findings raised the crucial question about the route(s) taken by *P. vivax* to colonize Latin America during the pre-Columbian period. It would have been unlikely that the parasite followed its human host across the Bering land bridge from Asia to Alaska given the absence of the mosquito vectors necessary for the completion of the parasite's life cycle at these high latitudes [10]. Nonetheless, the

unique ability of *P. vivax* to enter in a dormant state suggests an alternative scenario in which the parasite could have endured pre-Columbian transoceanic voyages from the Asian mainland or western Pacific islands to South America [11]. This hypothesis recently has gained support with the study by Rodrigues *et al.* [12] which documented genetic contributions from Melanesia (a group of islands in Oceania) in the mitochondrial gene pool of *P. vivax* American populations. Despite this evidence, studies using mitochondrial [12] or microsatellite markers [13] consistently indicated African and South-Asian populations as the main genetic sources of the present-day *P. vivax* lineages in Latin America. These studies also suggest the possibility of several independent *P. vivax* introductions into this continent during the transatlantic trade period [12,13].

More recently, the analysis of one *P. vivax* strain extracted from microscope slides from Spain, dating from 1942–1944, gathered as a composite sample called "Ebro", reignited the debate [14,15]. Indeed, this isolate, due to its genetic proximity to American populations, its position branching at the root of the American clade in the phylogenetic tree, and its estimated divergence coinciding with the transatlantic slave trade period led van Dorp *et al.* [15] to propose that *P. vivax* American populations originated from European colonization during the 15th century (*i.e.* in the post-Columbian period). This study introduced a compelling hypothesis of the European origin of *P. vivax* in Latin America. In 2024, Michel *et al.* [16] published four additional European *P. vivax* samples from the medieval to early modern period. Their findings further confirmed a strong genetic connection between European and Latin American *P. vivax* strains. Additionally, their study analyzed an ancient Peruvian sample dating from 1437 to 1617, which appears to support the presence of *P. vivax* in South America during the period of European colonization [16]. Nevertheless, both studies present some limitations. First, these studies did not consider all possible sources of American *P. vivax* populations. Indeed, each dataset included almost zero *P. vivax* individual from West or Central Africa (n = 0 in van Dorp *et al.* [15] and n = 1 in Michel *et al.* [16]). As more than seven million slaves, mainly from this region, arrived in Central and South America during the transatlantic trade [17], it is imperative to consider West and Central African *P. vivax* populations as potential sources of origin of *P. vivax* in Latin America. Moreover, an African origin must be taken into account because it has now been established that the colonization of Central and South America by *Plasmodium falciparum*, the most virulent human malaria agent, resulted from two independent introductions from West/Central Africa during the transatlantic slave trade [18–20]. The high frequencies of Duffy-negativity in African populations, expected to protect against *P. vivax* infection might exclude such a hypothesis; however, there is mounting evidence that *P. vivax* is present in West and Central Africa [21–23], as observed in Mauritania [24,25] and Senegal [26]. This reopens the door to this primary source. In addition, van Dorp *et al.* [15] only considered four populations from Latin America while Michel *et al.* [16] focused on just five populations from this continent, thus including only a limited number of populations and geographic areas within the parasite distribution range on the continent. Lastly, neither van Dorp *et al.* [15] nor Michel *et al.* [16] formally assessed the possibility of a admixed origin of American populations and/or multiple waves of *P. vivax* introduction into the continent.

Therefore, *P. vivax* evolutionary origin(s) in Latin America remains to be clarified and precised. The objective of this study was to perform a comprehensive population genomic analysis of a novel dataset of *P. vivax* genomes from multiple countries in Latin America and Africa. This study considered a large dataset included 1,133 isolates among which 214 were newly sequenced for this study. After quality filtering, the final dataset included 620 unrelated *P. vivax* genomes from 36 countries worldwide (Fig 1) among which, there were 107 newly sequenced genomes originating from six Latin American countries (Brazil, Ecuador, French

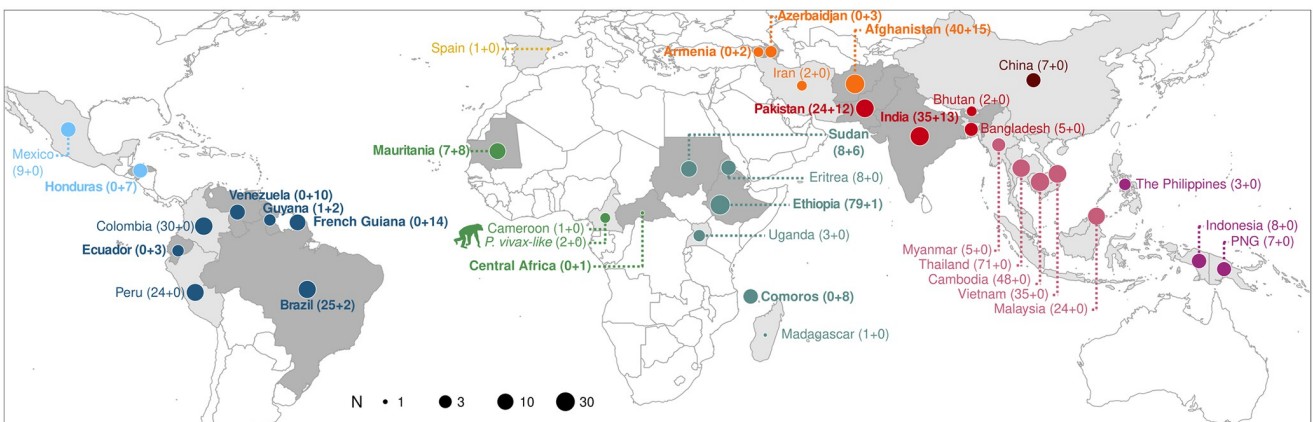

**Fig 1. Geographic origin of the 620 human *P. vivax* isolates and two African great apes *P. vivax-like* isolates.** Distribution of the samples studied per region: Central America (n = 16, light blue), South America (n = 111, dark blue), Europe (n = 1), West Africa (n = 18, light green), East Africa (n = 115, dark green), Middle East (n = 62, orange), South Asia (n = 91, light red), East Asia (n = 7, dark red), Southeast Asia (n = 183, pink), and Oceania (n = 18, purple). The circle size is proportional to the sample size in $\log_{10}$ units. The countries included in the sampling effort are differentiated by varying shades of gray.: darker gray for countries where new samples were sequenced and lighter gray for countries where already published samples were used. Country names highlighted in bold indicate that new genomes were sequenced in this study. In brackets, the number of samples from the literature is listed first, followed by the number of newly sequenced samples. The chimpanzee pictogram indicates the two great apes *P. vivax-like* samples studied. PNG: Papua New Guinea. The base layer of the map was made with Natural Earth (naturalearthdata.com).

Guiana, Guyana, Honduras, and Venezuela), five African countries (Central Africa, Comoros, Ethiopia, Mauritania, and Sudan), three Middle Eastern countries (Afghanistan, Armenia, and Azerbaijan), two South Asian countries (India, Pakistan), and one Oceanian country (Papua New Guinea). The combination of these new genomes with those already available provided a comprehensive assessment of the genomic diversity from a whole nuclear genome perspective by encompassing the present-day *P. vivax* distribution. To our knowledge, this study marks a new and original attempt to address the question of *P. vivax* colonization of Latin America using genetic material from nearly all modern potential source populations identified in the literature (Africa, Asia and Oceania) as well as the ancient "*Ebro*" sample from Europe. This dataset was used to study the population genetics structure and diversity, the population demographic histories, and to test different evolutionary scenarios of *P. vivax* colonization of Latin America using approximated Bayesian computations. Our results suggest a more complex *P. vivax* colonization history in Latin America than previously envisioned, with multiple waves of migration from Europe and West Africa. Our findings corroborate the European origin proposed earlier [15,16], but with a more recent colonization wave estimated during the post-colonial migration in the 19th century. However, they also emphasize the contribution of an additional unsampled population that might display a genetic affinity with West African *P. vivax* populations.

## Results

### A worldwide genomic sampling of *P. vivax*

In this study, we sequenced the whole genome of 214 new *P. vivax* strains from Central and South America, West, Central and East Africa, Middle East, South Asia, and Oceania (for more details see Materials and methods; S1 Table). Then, these new genomes were combined with 919 modern genomes from the literature [27–29] (see Materials and methods for more details; S1 Table and S1 Fig) and one ancient sample (Ebro) from Spain sequenced by van Dorp *et al.* [15]. After filtering out samples with >50% missing data (n = 68), those with multi-

strain infections (*i.e.*, multiple *P. vivax* genotypes, n = 320; S2 Fig), and those excessively related (n = 124), the final dataset included whole genome sequencing data of 619 modern *P. vivax* samples from 34 countries, one ancient DNA sample (Ebro) from Spain (Fig 1 and S1 Table) and two genomes of *P. vivax-like* from Cameroon, isolated from chimpanzees [28] and used here as outgroup. The mean sequencing depth ranged from 2.88 to 1,484.71 X for the modern genomes and the mean sequencing depth was 1.13 X for Ebro (S2 Table). The details about the bioinformatic pipelines, data processing, and the number of single nucleotide polymorphisms (SNPs) used are described in Materials and Methods and in S1 and S3 Figs. Some of our analyses considered the genotype likelihood using the ANGSD software ecosystem [30], rather than the classic SNP calling followed by hard quality filtering, to account for the genotyping uncertainty related to the sequencing depth heterogeneity. This strategy using ANGSD allowed considering 20,844,131 sites across the 620 genomes analyzed. Nevertheless, as some analyses required actual SNP calling followed by appropriate quality filtering, a Variant Call Format (VCF) dataset composed of 950,034 SNPs was also generated (see the Materials and methods and S1 and S3 Figs for more details).

## Genetic relationships of the American *P. vivax* with worldwide populations

To explore the genetic relationships of Latin American *P. vivax* isolates with worldwide *P. vivax* strains, the population structure patterns of 620 *P. vivax* isolates collected in 36 countries (Fig 1) were investigated using complementary population genomic approaches (see the Materials and methods and S1 Fig): (i) a principal component analysis (PCA) using *PCAngsd* [31], (ii) a model-based individual ancestry analysis using *PCAngsd* [31], and (iii) a maximum likelihood (ML) phylogenetic tree using *IQ-TREE* [32]. All three approaches showed consistent results splitting first the genetic variation geographically into four distinct genetic clusters (Fig 2): (1) Oceania, East and Southeast Asia, (2) Africa, (3) Middle East and South Asia, and (4) Latin America.

The first three principal components (PC) of the PCA (based on the genotype likelihood of 105,527 filtered and unlinked SNPs) revealed these four major genetic clusters (Figs 2a and S4). PC1 separated Oceania, East and Southeast Asia from Africa, the Middle East, South Asia, and Latin America. PC2 splitted American populations from all the others. African populations split from all the other populations on PC3. The same genetic structure with four genetic clusters was also recovered from the ML tree (Fig 2b). *P. vivax* populations from East and Southeast Asia, Bangladesh, and Oceania split first and cluster closer to the outgroup and apart from those from the rest of the world. Populations from Latin America, Middle East-South Asia and Africa clustered in three distinct lineages; with the last two being more closely related to each other than with the American lineages. Patterns of individual genetic ancestry (Figs 2c and S5) were also in line with results of the PCA and ML tree, displaying again four distinct genetic clusters. As the number of genetic clusters tested (K) increased, the four different genetic clusters also became distinct in terms of genetic ancestry. All the populations from East and Southeast Asia, Bangladesh and Oceania split first from the others at K = 2. Then, populations from Latin America split from Middle East-South Asia and from Africa, which became differentiated only at K = 4 (Fig 2c).

Data analysis at a finer sub-structuration level at K = 6 (the best K value based on the broken-stick eigenvalues plus one, following the recommendations of Meisner and Albrechsten [31]) (Figs 2c and S5) showed that Latin America remained as a single genetic cluster well separated from the rest of the world. Asia displayed three clusters (South Asia-Middle East, East Asia-Southeast Asia, and Oceania-Malaysia) and Africa was split into East and West Africa. Noteworthy, several American populations (Honduras, Ecuador, Brazil, Venezuela, French

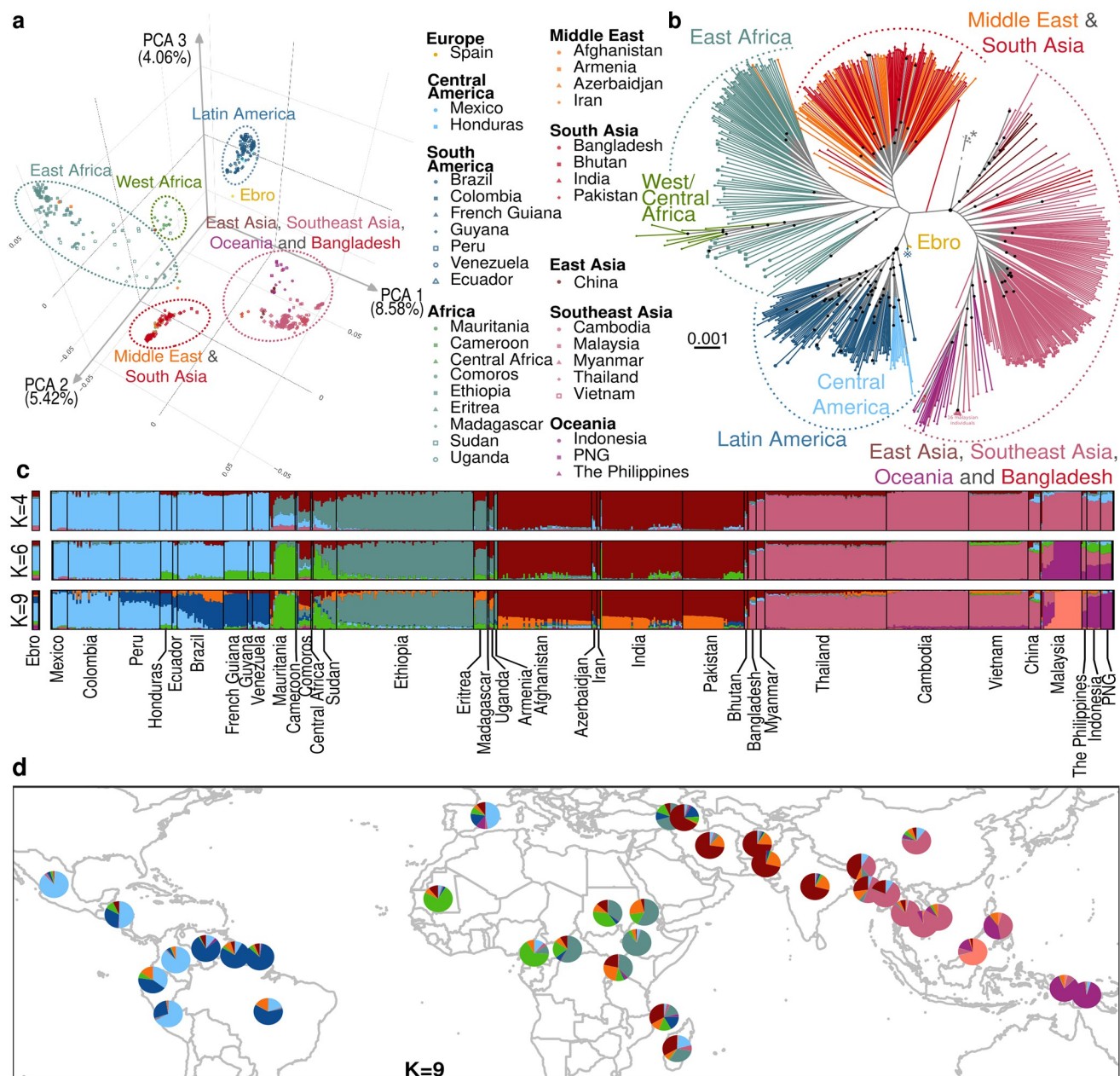

**Fig 2. Worldwide genetic structure of *P. vivax*.** (a) Principal component analysis of 619 modern *P. vivax* strains and the Ebro ancient DNA sample from Spain, showing the first three PCs based on the genotype likelihood of 105,527 unlinked SNPs (see S2 Fig for more details). (b) Maximum likelihood (ML) tree of the 620 *P. vivax* individual genomes obtained with IQ-TREE [32] based on a general time reversible model of nucleotide evolution [33], as determined by ModelFinder [34]. The ML tree includes two *P. vivax-like* strains from African great apes indicated by an asterisk (*) used to locate the root. Note that the length of the outgroup branch was truncated. Black dots at nodes correspond to highly supported nodes (SH-aLRT ≥ 80% and UFboot ≥ 95%). The reference mark (※) highlight a Brazilian *P. vivax* isolate, discussed in the text. (c) Individual genetic ancestry assuming K = 4 and K = 9 distinct genetic clusters estimated using *PCAngsd* (see S5 Fig for the other K values). (d) Geographic distribution of the population average genetic ancestry proportions at K = 9 estimated using *PCAngsd*. PNG: Papua New Guinea. The base layer of the map was made with Natural Earth (naturalearthdata.com).

Guiana, and Guyana) shared genetic ancestry with the cluster that contains West African isolates (in green, Fig 2c).

An additional sub-structuration among the American *P. vivax* populations was observed at K = 9 (Fig 2c). Specially, two distinct genetic clusters were identified, as previously observed [13,28,35]: (1) isolates from Mexico, Honduras, and Colombia gathered together as a Central American group, and (2) isolates from the French Guiana, Guyana, and Venezuela formed another distinct cluster as an Amazonian group (Fig 2d). The other populations from Brazil, Peru and Ecuador displayed patterns of admixed genetic ancestry between these genetic clusters.

In Africa, *P. vivax* populations were distributed in two distinct genetic clusters, one composed of the Mauritanian and Cameroonian samples, and the other including samples from Ethiopia and Eritrea. Sudan and Central African samples displayed admixed genetic ancestry between these African clusters (Fig 2d).

In Asia, a subset of the Malaysian population (n = 16) exhibited no evidence of admixture with other populations. These Malaysian samples were closely clustered in the ML tree (Fig 2b) and they were separated from the other Asian populations on PC 4 (S4 Fig). This pattern has already been reported before in this region [36] and it could be due to a strong bottleneck following a sharp decline in *P. vivax* prevalence in Malaysia over the past decade [37,38], which may have increased the population genetic drift.

In the PCA, the ancient DNA sample from Spain (Ebro) was genetically closer to the American populations than to any other population (Fig 2a), with which it shared a similar ancestry pattern (Fig 2c and 2d). In the ML tree, it clustered with one Brazilian sample (indicated with this symbol ※ in Fig 2b), the oldest sample from the Latin America dataset, collected in 1980 [29]. Similarly, in a previous study based on mitochondrial DNA, it clustered with South American samples [14]. Nevertheless, the nodes clustering these individuals were poorly supported in the ML tree (Fig 2b), preventing any strong interpretation about the origin of Ebro.

Overall, the population structure analyses (PCA, ancestry plots, and ML phylogenetic tree) supported the hypothesis that the Latin American *P. vivax* populations are genetically closer to the ancient DNA Ebro isolate from Spain, followed by the present-day populations from Africa and from Middle East & South Asia. Noteworthy, both the American and Spanish Ebro genomes displayed some admixed genetic ancestry with other populations from the rest of the world. For example, at K = 6 and 9 in the Admixture analysis (Fig 2c and 2d), Ebro shared approximately 50% of genetic ancestry with the American populations (light blue), but the rest originated from populations located in West Africa and Middle East (respectively light green and brown). Some American populations (Honduras, Ecuador, Brazil, Venezuela, French Guiana, and Guyana) also displayed shared genetic ancestry, particularly with isolates from West Africa.

## Population branching and admixture between American *P. vivax* and the rest of the world

To further investigate potential admixture events and identify candidate source populations that could have contributed to *P. vivax* colonization of Latin America, we first estimated population admixture graphs using two complementary approaches implemented in *TreeMix* [39] and in *AdmixtureBayes* [40].

The *TreeMix* analysis determined that the optimal number of migration edges was two or five (S6 Fig), while *AdmixtureBayes* analysis converged toward an optimal solution with two admixture events (S7 Fig). Both methods recovered the four genetic clusters (Fig 3) identified

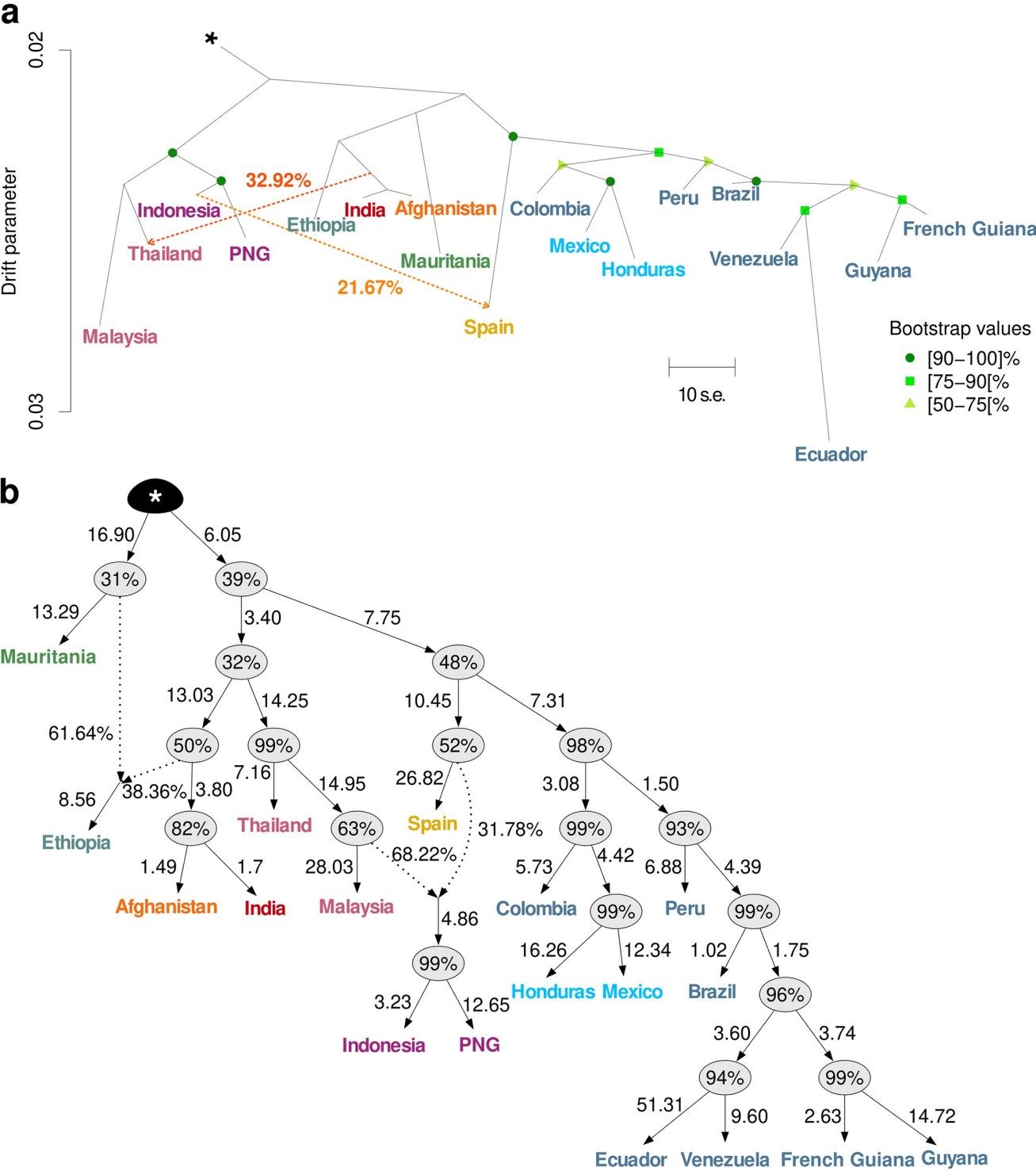

**Fig 3. Population graphs describing the genetic relationships and admixture proportions between *P. vivax* populations. (a)** Population tree estimated using *TreeMix* for a subset of 18 *P. vivax* populations with two migration edges (orange arrows) and rooted using the two *P. vivax-like* genomes indicated with the asterisk (*). The scale bar shows ten times the mean standard error (s.e.) of the covariance matrix. The migration weight is indicated as a percentage in orange on the arrows. **(b)** Network topology of a subset of 18 *P. vivax* populations with the highest posterior probability obtained with *AdmixtureBayes*, rooted with the two *P. vivax-like* genomes (*). The branch length indicates the genetic divergence between populations (measured by drift), multiplied by 100. The percentages in the nodes are the posterior probability that the true graph has a node with the same descendants. For each admixture event (indicated by the dotted arrows) the percentages illustrate the admixture proportion. PNG: Papua New Guinea.

by the genetic structure analyses (Fig 2): (1) Oceania, East and Southeast Asia, (2) Africa, (3) Middle East and South Asia, and (4) Latin America.

The American populations grouped into a single highly supported monophyletic cluster ($\geq$75% bootstrap support in *TreeMix*, Fig 3a, and $\geq$98% posterior probability in *Admixture-Bayes*, Fig 3b). This suggests that American populations most likely originated from a single introduction or from multiple introductions from genetically similar, but possibly admixed source populations from the rest of world. This second hypothesis is supported by the poorly resolved branching pattern close to the root in the *TreeMix* and *AdmixtureBayes* analyses (Fig 3). The branching patterns obtained in both population graphs showed a finer substructuration of the American populations (Fig 3): one cluster grouping Mexico, Colombia, and Honduras populations, another one with the French Guiana, Guyana, Venezuela and Ecuador populations, and in between, Brazil and Peru populations branched at an intermediate position reflecting their admixed genetic ancestry observed in the genetic structure analyses (Fig 2).

The ancient Spanish sample (Ebro) stood out as the isolate most closely related, but genetically distinct, from the American cluster, branching at its base. This ancient isolate had a notable long branch that reflected important genetic drift compared with other present-day populations. Such a long branch may be an artifact because it is a single haploid sample from an ancient population [41]. As rare and common alleles in that ancient population can not be properly estimated, this usually leads to an inflated estimate of the genetic drift as previously reported [41]. Additionally, the *TreeMix* and *AdmixtureBayes* analyses detected an admixture event between Ebro and Oceanian populations, corroborating the shared genetic ancestry identified in the population genetic structure analyses (Fig 2d and 2c). The *TreeMix* analysis also identified a migration edge from the common ancestor of India and Afghanistan into Thailand.

To assess further potential admixture events between the American populations and those from the rest of the world, the admixture $f_3$-statistics were computed. This involved testing American populations against potential source populations from both within Latin America and from all other geographical regions worldwide. None of the tested $f_3$ combinations yielded a significantly negative result, as expected if admixtures occurred between the American populations and one of the two tested sources (S8 Fig). As underlined by the authors of the $f_3$ admixture test [42], these non-significant results do not preclude that admixture could have occurred in a distant past, with a residual signal remaining only in a minor fraction of the genome. This would be also expected if the source population(s) that founded the Latin American populations were already admixed.

## Effective population size changes and split time of American *P. vivax* populations

The next step was to determine whether colonization of Latin America by *P. vivax* was associated with founder effects and population bottlenecks, as can be expected when colonizing a new environment. To this aim, the nucleotide diversity ($\pi$) as well as the Tajima's D values [43] were computed in the core genome, considering only populations with at least 10 samples. The $\pi$ values in the American populations (S9 Fig) were marginally lower compared with those observed in other populations, with exceptions in Malaysia and Ethiopia. In Latin America, the $\pi$ values displayed distinct patterns: populations along the Pacific coast (Colombia and Peru) exhibited significantly lower median $\pi$ values than populations in the Amazon (Brazil, French Guiana, and Venezuela) ($\sim 8 \times 10^{-4}$ versus $\sim 1 \times 10^{-3}$, *p-value* $<0.001$, Wilcoxon signed-rank test and Bonferroni correction, S9 Fig). The Tajima's D distributions in the same *P. vivax*

populations were largely negative, indicating excess of rare variants over shared variants, consistent with demographic expansions. However, the Tajima's D distribution of populations from Colombia, Peru, and Ethiopia were closer to zero, with a notable presence of positive values (S9 Fig). Overall, these descriptive statistics are consistent with the demographic expansion of the American populations. They also suggest that no strong bottleneck occurred during *P. vivax* invasion history of Latin America or that the founding isolates were somewhat admixed and harbored genetic diversity comparable to that of the source populations.

The changes in effective population size ($N_e$) with time were inferred using the coalescence rates (CR) estimated with the software *Relate* [44] for modern samples. Consistently with the π and Tajima's D values, these analyses showed that all modern populations across the world displayed similar trends, characterized by a monotonic decline in $Ne$ values that started roughly ~100,000 generations ago (~20,000 years ago, assuming a generation time of 5.5 generations/year [28] and a mutation rate of 6.43 x $10^{-9}$ mutations/site/generation [28]) followed by a recent and moderate expansion (Fig 4a). The recent growth observed in *P. vivax* populations around the world might be due to historical bottlenecks, possibly caused by climate changes, human migrations, or development of host immunity. Earlier studies have shown that *P. vivax* populations expanded after these bottleneck events [45,46].

Noteworthy, all American *P. vivax* populations displayed slightly lower $Ne$ values than the other populations, and only three populations from Africa and Oceania showed comparable trends (Mauritania, Malaysia, and Papua New Guinea). The $Ne$ values of all American populations became noticeably distinct from the other populations since the last ~500 to 2,000 generations (~100–300 years ago). This time range suggests a putative divergence time of the American populations from the other populations, but also with each other. Besides the trees (Figs 2b and 3) where the American populations formed a single well-supported cluster, the synchronous departures of the $Ne$ values of the American populations, compared with those from other populations worldwide, also suggests a single introduction from one source or multiple introductions from genetically similar and possibly admixed sources in Latin America.

The divergence time estimates were refined using the inverse cross-coalescent rate (ICCR) between representative populations from the American clusters (Colombia and French Guiana) and other populations in the world that are representative of distinct genetic clusters (Figs 2 and 3), as well as the ancient isolate from the Ebro region in Spain. This analysis used the program *Relate* [44] for modern samples (Fig 4b) and the program *Colate* [47] for the ancient Ebro sample (Fig 4c). The split time between populations can be estimated by comparing the changes in ICCR and ICR values between populations and within population respectively. The split time corresponds to the time when the ICR values within each population depart from each other, and the ICCR values between populations increase. This means that the CR between populations decreases, and therefore they are diverging [48]. According to the ICCR variation (Fig 4b), the divergence times estimated between the American populations and Asian (*i.e.* Thailand) or Oceanian (*i.e.* PNG) populations occurred more than 1,000 ybp (or more than 55,000 generations ago). The divergence time between populations from West Africa and Latin America dated back 500–1,000 ybp (~3,000–5,500 generations ago). Lastly, the split between the American populations and the ancient Ebro isolate from Spain would have been more recent (100–300 ybp or 550–1,650 generations ago).

## The role of West African and European *P. vivax* populations in the invasion of Latin America

Our analyses of the population structure and relationships between populations indicated that American *P. vivax* populations form a highly supported monophyletic cluster (Figs 2b and 3).

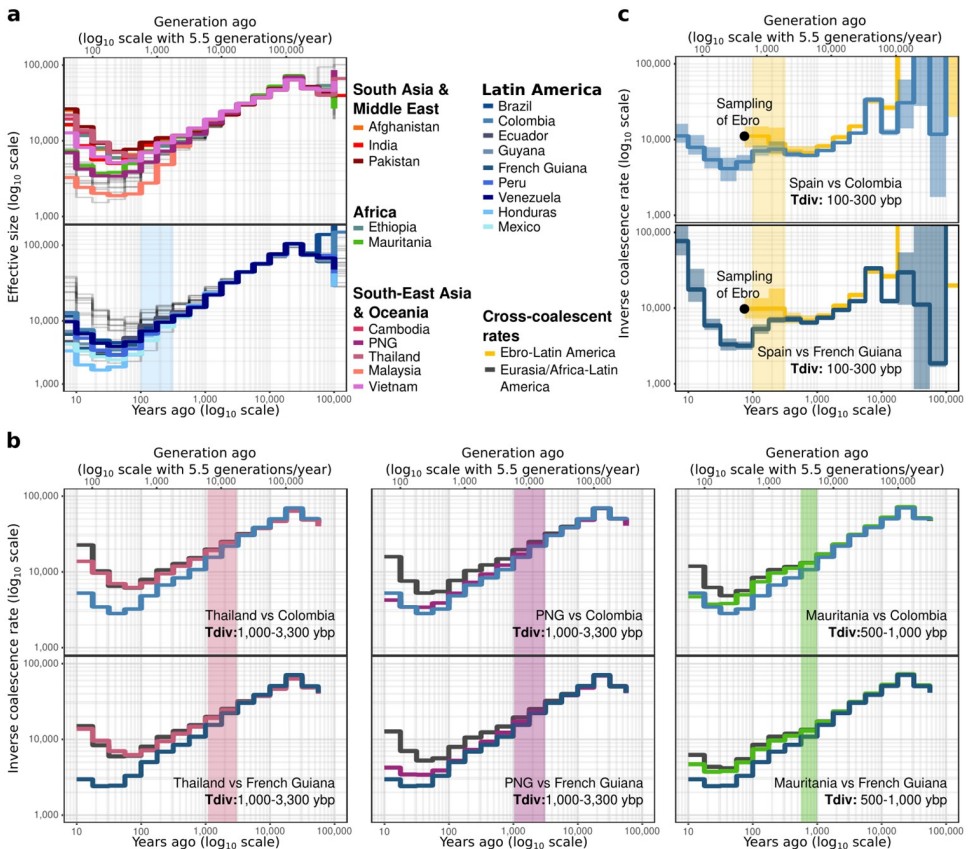

**Fig 4. Coalescent-based inference of the demographic history of *P. vivax* populations. (a)** The variation in effective population ($N_e$) size was estimated using *Relate* (axes are $log_{10}$ transformed). The period highlighted by the blue rectangle corresponds to the diversification of the Latin American populations. **(b)** Comparison between the inverse coalescence rates (ICR) and inverse cross-coalescence rates (ICCR) from *Relate* between Latin America (Colombia on top, French Guiana below) and Eurasia/Africa (left to right: Thailand, Papua-New-Guinea, Mauritania) provides a formal estimate of their split time [46]. The axes are $log_{10}$ transformed. The periods highlighted by the colored rectangles correspond to the divergence time (*Tdiv*) between populations and are also specified in the panels. **(c)** Comparison of the ICR and ICCR between Latin America (Colombia on top, French Guiana below) and Ebro. The 95% confidence interval resulting from 100 bootstraps inferred using *Colate* is indicated by the envelope surrounding the estimate line. The black dot indicates Ebro sampling time. Axes are in $log_{10}$ scale units. The period highlighted by the light-yellow rectangles corresponds to the divergence time (*Tdiv*) between the American populations and Ebro and is specified. PNG: Papua New Guinea.

These results and the concomitant divergence of all American populations (Fig 4a) suggested that they descend from a single introduction from one source or from multiple introductions from genetically similar and possibly admixed sources. The ancient Ebro sample from Spain was the isolate most closely related to the American populations. It displayed an admixed genetic ancestry, half of which was shared with the American *P. vivax* populations, and the other half originated from multiple sources (Africa, the Middle East, and even Asia) (Fig 2c). Nevertheless, West African *P. vivax* populations may have also contributed to the genetic make-up of the Latin American populations. This is suggested by the recent human history that involved massive population movements between Europe, West Africa and Latin America [17]. This is also supported by the genetic ancestry analyses showing that to some extent, West African populations have contributed to the genetic ancestry of *P. vivax* populations in Latin America (Figs 2c and S5). In the literature, analyses using mitochondrial DNA [12] or

microsatellites [13] similarly indicate a likely West African influence, introduced during the transatlantic slave trade. However, no evidence of significant admixture was detected in the population graph analyses between American and African populations (Figs 3, S6 and S8). Furthermore, American populations did not share ancestry with Asian or with Oceanian populations (Fig 2c), and the divergence times with Asian and Oceanian populations were much older than with West African or European populations (Fig 4). The European ancient Ebro isolate and the West African populations were the only groups from Eurasia and Africa that showed credible evidence of shared genetic ancestry with the American populations. Therefore, it was important to test formally their putative roles as source populations in the *P. vivax* invasion history of Latin America.

We used the simulation-based approximate Bayesian computation (ABC) statistical framework that relies on the supervised machine learning Random Forest tree classifier (ABC-RF), as implemented in DIYABC-RF [49]. This approach allowed formal statistical comparisons of distinct *P. vivax* invasion history scenarios of Latin America that differed in terms of population branching order, admixture events, time of split and admixture, and that involved or not unsampled (or ghost) populations (see Fig 5a and Material and method for details). In an ABC-RF analysis, different demographic scenarios (see parameters in S3 Table) are tested by simulating genetic data under each scenario. Then, summary statistics from the simulated data under each scenario are computed and used to train the ABC-RF tree classifier to assess which scenario best fit with the observed data [49,50]. This analysis considered twelve colonization scenarios that involved three representative populations: one from West Africa (Mauritania), one from Europe (the Ebro sample), and one from America (either Colombia or French Guiana). Colombia and French Guiana were selected as representative populations because they had the largest non-admixed sample sizes within each Latin American subgroup (n = 30 for Colombia and n = 14 for French Guiana). To determine whether these subgroups shared the same invasion history in Latin America, we conducted the analysis twice, each time considering one of the two American populations as representative. An "ghost" population was also introduced in scenarios 5 to 9, 11, and 12 to assess the potential contribution of a unsampled population, which can be also extinct (Fig 5a). Scenario 2 considered only a European origin; scenarios 3, 4 and 12 considered only an African origin; and scenarios 1, 6, 7, 8, 9, 10 and 11 considered various combinations of admixed histories.

In both analyses, whether considering the Colombian or French Guianese American representative population, the most likely scenario identified was scenario 6 (Fig 5). This scenario suggests that the American *P. vivax* populations originated from an admixture event between the historical European population and an unsampled population derived from an ancestor to the European gene pool that branched with West African ancestral populations. This scenario received the highest proportions of ABC-RF classifier votes and this result was consistent over 10 independent replicate analyses (mean RF vote proportions ± standard deviation = 18.4 ± 1.0% with Colombia and 21.90 ± 1.06% with French Guiana) (Fig 5a and S4 Table). The mean posterior probability for this best supported scenario was 46.2 ± 1.3%. After scenario 6, the two other best scenarios supported by the ABC-RF analysis were Scenarios 4 and 8 with Columbia (mean RF vote proportions = 14.4 ± 0.5% and 13.2 ± 0.9%, respectively) and scenario 8 and 11 with the French Guianese American representative population (mean RF vote proportions = 15.03 ± 1.02% and 12.21 ± 0.94%). Scenario 4 assumes that Ebro could be a migrant from Latin America, while American populations would descend from an ancestral population of West African origin. In scenario 8, the American populations would have derived from an admixture event between the European populations and an unsampled population of direct West African origin. Thisscenario is actually close to scenario 6. Combined, the three best scenarios (6, 4 and 8) identified with Columbia collected 46.0% of all the votes, while each of the

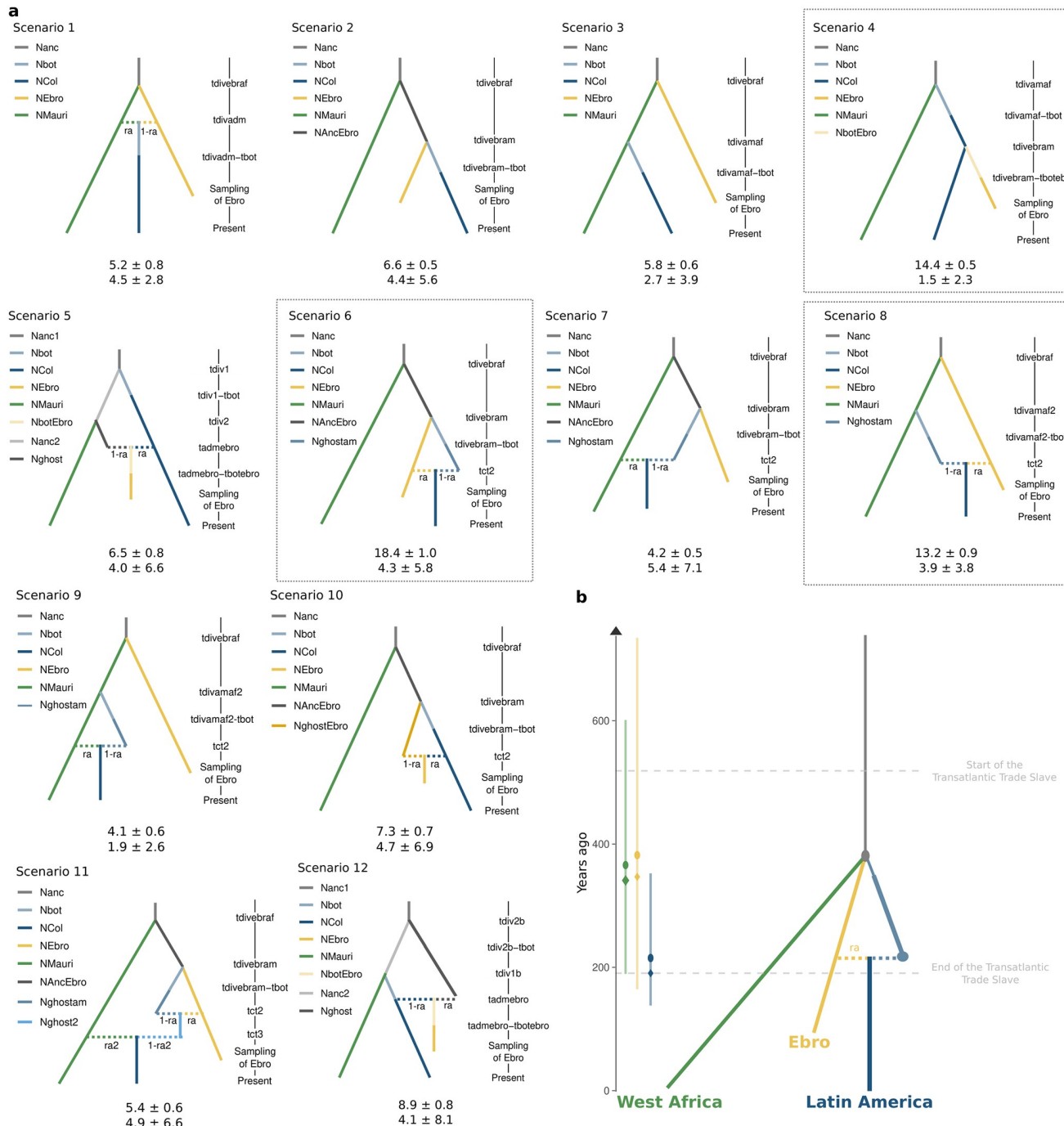

**Fig 5. Most likely *P. vivax* colonization history scenarios of Latin America inferred using the Approximate Bayesian Computation Random Forest approach. (a)** Twelve colonization scenarios were tested. Distinct solid colored lines correspond to populations with distinct effective size parameters (labels starting with "N") and horizontal dashed lines represent admixture events with a contribution *ra* and 1-*ra* from each of the two contributors. Time parameters of the different events are displayed along the vertical line on the right of each scenario (labels starting with "t", not to scale). The parameters of each scenario are characterized by probabilistic distributions detailed in S3 Table. Under each scenario are the mean percentage of RF classification votes ± standard deviation (SD) and the mean type II error ± SD. The three most supported scenarios (scenarios 6, 4 and 8, with 46.0% of the total votes) are highlighted by dotted-line rectangles. **(b)** The best-supported scenario (scenario 6) in the DIYABC-RF analysis, scaled to relative time-parameter estimates (converted to years assuming a generation time of 5.5 generations/year). On the left, each time parameter estimate is indicated by the mean (circle), median (diamond), and 90% confidence intervals (colored bars). Here, the results are only shown considering the Colombian population as a Latin America representative population. Results considering the French Guiana population were very similar (see S4 Table).

other nine scenarios received less than 8.9% of all votes (Fig 5a). Scenario 11 identified as the third best scenario when considering French Guiana was actually a more complex version of scenario 6 and 8 combined together. It assumed that the American populations could result from two successive admixture events, the first and oldest one between the European populations and an unsampled ghost population, followed by a more recent wave of introduction from West Africa (Fig 5a). The combined RF-votes for these three best scenarios (6, 8 and 11) identified with the French Guinanese population was 49.14%(S4 Table). In both analyses, ten replicates of the ABC-RF analysis supported strongly and unanimously this result. Furthermore, the confusion matrix (S10 Fig) and the prior error rate of 42.1 ± 0.0% indicated the overall performance of the RF classification analysis and highlighted high power to distinguish among the twelve competing scenarios.

Model parameter estimates for the optimal scenario 6 indicated that the average divergence time between the ancestral European (Ebro) population and the West African population (in green in Fig 5b) was ~350 years ago (or ~2,000 generations ago). Similarly, the average estimated divergence time between the unsampled (ghost) population and Ebro (in yellow in Fig 5b) was ~400 years ago (or ~2,100 generations ago). Therefore, the ABC-RF analysis suggested that the two split times between West Africa, Europe and the unsampled population were nearly synchronous. These split time estimates from the ABC-RF are also very close to those from the CCR estimates obtained using *Relate* and *Colate* between the American populations and those from Europe (Ebro) and West Africa (Fig 4). Furthermore, the admixture event between the unsampled population and the European population (in yellow in Fig 5b) most likely occurred ~200 years ago (or ~1,200 generations ago), with a predominant European contribution ($ra$ = 60%).

Altogether, these ABC-RF results support the hypothesis that American *P. vivax* populations descended from now-extinct European populations, but also from unsampled population(s), possibly originating from West Africa, an event that happened during the post-colonization migration of the American continent (second half of the 19th century).

## Discussion

The investigation of the colonization of Latin America by *P. vivax* has been approached through a variety of methodologies. Previous studies using mitochondrial DNA [12] and immunohistochemistry [5] suggested the existence of a pre-Columbian presence and potential Asian/Oceanian origins. Other studies using mitochondrial DNA [12] or microsatellites [13] indicated a potential influence from West Africa during the transatlantic slave trade era. More recently, the exploitation of nuclear genomes from five ancient European *P. vivax* strains suggested a European introduction that coincided with the European colonization of Latin America [15,16]. However, these studies had important sampling limitations, particularly the absence of populations from West Africa, the region from where millions of enslaved persons were transported to Latin America during the transatlantic slave trade [17] and the source of American *P. falciparum*, another human malaria parasite [18–20].

Considering these factors, population genomic analyses were conducted using a novel dataset of 620 *P. vivax* isolates from 36 countries, including 107 newly sequenced samples, particularly from Latin America and West Africa. To the best of our knowledge, this dataset represents the first comprehensive collection of nearly all potential source populations proposed in the literature, thereby allowing the exploration of the evolutionary history of *P. vivax* colonization in Latin America. In addition to population structure and demographic history analyses, the influence of European and African populations in the colonization of the

American continent was tested by comparing various colonization scenarios using a supervised machine learning ABC-RF approach [49,50].

## *Plasmodium vivax* genetic structure and diversity in Latin America

The analysis based on PCA, ancestry plots, ML phylogenetic tree, and population admixture graphs revealed a consistent genetic structure among the Latin American *P. vivax* populations within the Americas and as well as to *P. vivax* populations worldwide, as previously described [29,36,49]. As previously reported [13,28,35], in Latin America, two separate genetic clusters were identified, representing the Central American region (Mexico, Honduras, and Colombia) and the Amazonian region (French Guiana, Guyana, and Venezuela). Genetic structure analyses also indicated the presence of an admixed cluster, thus suggesting potential gene flows between these clusters in Latin America. Several studies confirm cross-border spread between South American countries [13,35,51]. These results highlight the possibility of gene flows between Peru, Brazil, Guyana, Venezuela, and French Guiana that could be explained by the large asymptomatic human reservoirs and the dormant stage characteristic of *P. vivax* [52–54].

The geographic distribution of genetic diversity across the Latin American region revealed that *P. vivax* populations along the Pacific coast (Colombia and Peru) exhibited lower genetic diversity ($\pi$) compared with Amazonian populations (Brazil, French Guiana, and Venezuela) (S9 Fig). This is in line with previous studies [35,55], and could stem from transmission pattern variations across regions [35]. Indeed, coastal areas experience more periodic malaria transmission events compared with the Amazon region, where transmission is more stable throughout the year [56,57]. However, some studies found that *P. vivax* populations can exhibit high genetic diversity during periods of low transmission rates [55,58]. This means that data on spatial variation in genetic diversity must be interpreted with caution.

## Shared genetic ancestry between *P. vivax* from Latin America, Europe, and West Africa

The population genetic structure analyses (PCA, ancestry plots, and ML phylogenetic tree) showed that *P. vivax* populations from Latin America are genetically closer to the ancient DNA Ebro isolate from Spain, followed by present-day populations from Africa, Middle East and South Asia (Fig 2). Noteworthy, both the American isolates and the Ebro isolate displayed some admixed genetic ancestry with other populations from the rest of the world. At first glance, this may suggest that the European *P. vivax* may have been the only genetic source involved in the colonization of the American continent, as suggested in some studies [15,16]. Nevertheless, this does not rule out the possibility that other populations may have also contributed, especially since various demographic histories may result in similar admixture patterns [59]. When considering the admixture analysis, >50% of the Ebro genetic ancestry was shared with the American populations (light and dark blue), but the rest originated from West African and Middle Eastern populations (light green and brown, Fig 2c and 2d). In addition, some American populations (Honduras, Ecuador, Brazil, Venezuela, French Guiana, and Guyana) also displayed shared genetic ancestry with those from West Africa. These results support European participation in the genetic make-up of Latin American *P. vivax* populations, but also indicate that the history of the invasion of Latin America could have been more complex with the potential involvement of West African genetic sources.

## No genetic evidence of an Asian contribution to the Latin American *P. vivax* populations

Previous studies suggested that Asian *P. vivax* populations might have contributed to the colonization history of Latin America. For instance, a study based on mitochondrial genomes found evidence of genetic ancestry from Oceanian *P. vivax* strains in American populations [12]. Our genomic analysis also suggested a potential introgression from Indonesia to Mexico, as shown by the *TreeMix* results when considering up to m = 5 migration edges (the second optimum, see S6 Fig). However, this introgression appears to be minimal, close to 0% (S6 Fig), and was not corroborated by the $f_3$-statistics (S8 Fig). This introgression could have been the consequence of some historical introductions of Asian *P. vivax* during human migrations, but it could also mirror part of the history of the European *P. vivax* populations, that were the main sources of American *P. vivax*. Indeed, the population graph analysis of *TreeMix* and *AdmixtureBayes* (Fig 3) detected admixture events that involved Ebro and the Oceanian populations (Papua New Guinea and Indonesia). These admixture events may be due to genetic exchanges between European and local parasites from the Philippines, an archipelago of islands close to Indonesia that was a Spanish colony for more than three centuries (from 1565 to 1899). The reported contribution of Melanesian *P. vivax* isolates to American *P. vivax* genomes may reflect historical introgression, admixture events, or incomplete lineage sorting within the Spanish genomes. Therefore, the likelihood of Asian *P. vivax* serving as genetic sources in the colonization of Latin America appears relatively low, and possibly also integrated within the history of European populations.

## Latin American *P. vivax* populations shared genetic ancestry with the now-extinct European population, but not only

As the Ebro sample showed closer genetic ties with American *P. vivax* populations than with Middle East populations (Figs 2 and 3), several scenarios where Ebro came from Latin America (scenarios 4, 5, 10 and 12 in Fig 5a) and scenarios where Ebro contributed to the genetic make-up of Latin America (scenarios 1, 6, 8 and 11 in Fig 5a) were tested. The best scenario (scenario 6: 18.4% of votes) suggested that European *P. vivax* populations are one of the main sources of the genetic make-up of American *P. vivax*. Conversely, the second best scenario (scenario 4: 14.4% of votes) suggested that Ebro could be a migrant from Latin America (Fig 5a). This hypothesis could be rejected because Michel *et al.* [16] recently showed that Ebro is genetically part of the European cluster, composed of samples that predate Christopher Columbus's arrival in the Americas.

Our ABC-RF scenario testing analysis supported the major contribution of European populations in shaping the genetic ancestry of Latin American *P. vivax* populations. Yet, these results also suggest that the colonization history of the Latin American continent might have been more complicated. Specifically, our analysis indicates a potential contribution from West African populations and/or from an unsampled population to the origins of *P. vivax* in Latin America. According to the best supported scenario (scenario 6 in Fig 5), the West African, Ebro and unsampled populations would have all split almost at the same time. Then, the admixture between the European population and the unsampled population would have composed the Latin American populations. To ascertain the role of West African *P. vivax* as a genetic source in the colonization of Latin America, it is imperative to include additional European *P. vivax* samples from ancient specimens, as the ones sequenced by Michel *et al.* [16] in future studies, and also to increase the African sample size and its distribution. Indeed, samples from central African countries such as Angola, where one of the main ports of the transatlantic trade was located [60], could provide valuable insights into the evolutionary history of *P.*

*vivax*. However, at the time of this study, such samples were not available. Furthermore, more elaborated simulation frameworks using demographic-genetic scenarios including gene flow (and not just admixture events) and considering dormancy may bring new insights into the complicated *P. vivax* colonization history of Latin America.

## Distinct invasion waves of *P. vivax* in Latin America

The results obtained in this study showed that European, West African, and unsampled populations all diverged at the same time, about 400 years ago (Fig 5b). However, *P. vivax* was probably already present in Latin America at this time, as demonstrated by Michel *et al.* [16]. This study reported a sample of *P. vivax* that clusters with modern Peruvian *P. vivax* strains. This ancient strain was sampled at Laguna de los Cóndores (Peru) and dated between 1437 and 1617, whose host is thought to be a person of Amerindian descent. Thus, the presence of this ancient American sample suggests that the unsampled population came from a first wave of invasion in Latin America, at least before the 17th century, and probably during European colonization in the 15th century.

The second significant event in the *P. vivax* colonization scenario identified by the DIYABC-RF is the admixture between the European population and the unsampled population in which the European population contributed approximately 60% of the current American genetic pool. Our analyses suggest that this event occurred towards the end of the transatlantic slave trade, in the latter half of the 19th century. This time frame coincides with the massive post-colonial migrations [61], when more than 10 million Europeans settled in Latin America [62], potentially introducing parasites in this continent.

The presence of *P. vivax* before the 17th century in Latin America [16] and the results presented in this study, therefore suggest that there were at least two waves of *P. vivax* introduction into Latin America: a first wave prior to the 17th century, then a second from Europe at the end of the 19th century. There is no way to determine from where, following divergence, this first wave originated. It could be from West Africa. This scenario is partially supported by the observed shared ancestry with some American populations (Fig 2c) and our DIYABC-RF analyses. Indeed, in scenarios 4, 8 and 11 (the second and third most supported scenarios in the DIYABC-RF analysis with Colombia and French Guiana as representative American population) (Fig 5a), American populations came from West Africa or from a population descended from West Africa and a European population. Currently, the only way to identify the exact genetic origin of this unsampled population is to generate more genetic data from these regions and/or using ancient samples (*e.g.* historical slides, historical mosquito specimens).

Noteworthy, our divergence time estimates between Europe and Latin America are approximately 400 years younger than those estimated by van Dorp *et al.* [15]. Such a difference may be explained by the different methodologies used in the two studies. We used a generation time of 5.5 generations per year [28], but other generation times have been suggested [63] and can also vary depending on the region [64], thus affecting the estimates of divergence times in years. van Dorp *et al.* [15] did not use any generation time estimate and inferred the divergence times by relying on the correlation between collection times and root-to-tip distances of the different samples [65]. This correlation was calibrated using only two historical points: the Ebro composite sample from the 1940s and a North Korean sample from 1953. Furthermore, the fact that Ebro is a composite sample may bias the divergence time estimate. Indeed, the samples in Ebro were collected between 1942 and 1944 and this was not taken into account in the correlation. Moreover, the mutations observed correspond to several isolates, and not to a single one. This may lead to a higher divergence time due to the artificial accumulation of

mutations. Unlike van Dorp *et al.* [15], the two analyses used in our study converged toward very similar estimates of the divergence time between Ebro and American populations. Indeed, *Relate* gave a divergence time of 100–300 ybp (Fig 4c) and DYABC-RF suggested a divergence time of 212 ybp (Fig 5b).

Estimates on the effective population size and divergence time in *Relate* rely on a choice of mutation rate, which is alway difficult to make. Here we chose a value commonly used in the literature [28], but we acknowledge that mutation rate can vary both spatially and temporally in *P. vivax*. We expect that it will bring uncertainty on the absolute value, but not relative to each other. Furthermore, DIYABC-RF does not require making assumptions about the mutation rates, since simulations are done considering parameters values that are scaled by the mutation rate, following the Hudson simulation framework for SNPs [66]. Therefore, DIYABC-RF provides marginal parameter estimates that are scaled by the very high uncertainty of the mutation rate [49,50]. Usually, this leads to very high uncertainty as observed in Fig 5b, but the fact that both *Relate* and DIYABC-RF methods converged toward very similar estimates of divergence times suggest the mutation rate we use may be a reasonable approximation for *P. vivax*.

The approaches used by van Dorp *et al.* [15] and in the present study both rely on the Wright-Fisher model for coalescent simulation (*a.k.a.* the Kingman's coalescent model [48]), which assumes a panmictic population with non-overlapping generations and neutral evolution [67,68]. However, in the case of *P. vivax*, the assumption of non-overlapping generation is not met, particularly because of dormancy. Especially since *P. vivax* emerges from dormancy when the host is infected with new strains [3,69]. As a result, diversity is maintained through generation, which may reduce the effect of genetic drift and lead to somewhat overestimated effective population size [58,70]. This may also explain why the Tajima's D values (S9 Fig) and population size inference of the American populations (Fig 4a) did not suggest any bottleneck resulting from colonization, as previously observed [55]. Furthermore, the Kingman's coalescent model allows only two lineages to merge at each generation. This also departs from *P. vivax*'s life cycle. Its alternation between asexual reproduction in humans and sexual reproduction in mosquitoes can lead to multiple simultaneous coalescent events in a generation [71]. Simulations by Korfmann *et al.* [72] show that using Kingman's model underestimates effective population sizes when such events are frequent. However, Kingman-based tools remain accurate if these events are rare [72]. To mitigate this bias, we removed clonal and related individuals (see Materials and methods), but we cannot rule out that some bias may remain.

Additionally, our analysis overlooked migrations between populations, that can influence the estimates of genetic parameters, such as divergence times and population sizes. Omitting migration and overlapping generation can reduce the divergence times estimates and increase the inferred population sizes. Linked selection can also lead to underestimated the effective population size and divergence times [73], although we reduce this bias by using unlinked SNPs for these analyses (see Materials and methods). These factors could also explain why the trifurcation of Europe, West Africa, and the unsampled population is more recent than the earliest *P. vivax* presence detected in Europe and Latin America [16]. However, this trifurcation may also indicate that the transatlantic trade (from the 15[th] to the 19[th] centuries) facilitated contact between the European and West African *P. vivax* populations. Incorporating *P. vivax*-specific life-history and epidemiologic traits and selection in the population genetic inferences could improve their accuracy, but this currently represents major challenges. This would require theoretical and simulation-based population genetic studies, possibly using heavy forward genetic simulations such as those implemented in *SLiM* [74] and complex and heavy parametrization on each stage of the *P. vivax* life cycle, as well as on its temporal and spatial variations.

## Conclusions

This study reveals that *P. vivax* invasion of Latin America was not exclusively a result of European colonization during the 16[th] century. Indeed, our results indicate a significant genetic flow during the second half of the 19[th] century, primarily driven by European post-colonial migration. However, Europe would not be the only source of *P. vivax* in Latin America: an additional, unsampled population likely contributed to the parasite genetic diversity in this region, either during early contact between the Americas and Europe or via the transatlantic slave trade. Future studies should focus on obtaining *P. vivax* samples from different parts of the world (Middle East, Central and West Africa) to identify this population, including also historical samples from Europe and North America, such as those recently published [15,16], but from other regions as well, such as the Caribbean and Central Africa.

It is important to note that significant human migrations played a pivotal role in the invasion of the Americas by the two main human malaria agents, *P. falciparum* and *P. vivax*. *P. falciparum* entered the continent with the millions of African slaves who were brought to the Americas from the 16[th] to the 19[th] century [18,19], and *P. vivax* might have accompanied millions of European migrants in the late 19[th] century.

It is now imperative to investigate traces of selection in American *P. vivax* genomes. Understanding the genetic basis of *P. vivax* colonization success in the Americas is crucial for developing effective control strategies and predicting invasions of other regions. Yet, the current literature lacks comprehensive insights into this aspect [75], highlighting the need for more studies in this field.

## Materials and methods

### Ethics statement

In this study, 215 new *P. vivax* isolates from 24 countries were sequenced (Fig 1). *P. vivax* infection was detected by microscopy analysis, polymerase chain reaction (PCR) amplification of the *Cytochrome b* gene, and/or rapid diagnostic testing. Samples were obtained from patients infected with *P. vivax* following their informed consent and approval from the local institutional review board in each country. The informed consent process for the study involved presenting the study objectives to the community and inviting adults to participate. Before sample collection, each individual was informed about the study purpose and design and provided with a study information sheet. Oral informed consent was obtained from each participant. As the study posed no harm, it did not require written consent for its procedures. This oral consent approach aligned with the ethical standards of each country at the time of enrollment and was endorsed by the local ethics committees. Additionally, measures were taken to ensure the privacy and confidentiality of the collected data through sample anonymization before the study started.

For samples from Honduras, ethical and scientific approvals were obtained from the Ethics Review Committee of the Infectious and Zoonotic Diseases Master's Program at Universidad Nacional Autónoma de Honduras (CEI-MEIZ 02–2014; 5/19/2014). Data were analyzed anonymously because the study made secondary use of biological specimens originally collected for malaria diagnosis as per the standard of care in Honduras. For samples collected in Venezuela, each patient gave a written informed consent, and ethical clearance was obtained from the Comité Ético Científico del Instituto de Medicina Tropical de la Universidad Central de Venezuela. Ecuadorian samples were collected through the Ministry of Health malaria surveillance program. The protocol was approved by the Ethical Review Committee of Pontificia Universidad Católica del Ecuador (Approval Number: CBE-016-2013). Informed written consent was

obtained from all participants and/or their legal guardians in the case of minors before sample collection. For four samples from Mauritania, the study was approved by the pediatric services of the National Hospital, the Cheikh Zayed Hospital, and the Direction régionale à l'Action Sanitaire de Nouakchott (DRAS)/Ministry of Health in Mauritania. No ethics approval number was obtained at this time. For samples from Azerbaijan and Turkey, *P. vivax* was isolated from patients as part of the routine primary diagnosis and post-treatment follow-up, without unnecessary invasive procedures. The informed consent of each patient or an adult guardian of children enrolled in this study was formally obtained at the moment of blood collection. Written informed consent was collected, which included clear information that the samples would be used to investigate the genetic diversity of Plasmodium parasites as part of the VIVA-NIS project, supported by the COPERNICUS-2 RTD project contract ICA2-CT-2000-10046 of the European Commission. For child participants, formal written consent was obtained from a parent or legal guardian before sample collection. For the remaining newly sequenced samples, no specific consent was required because, in coordination with the Santé Publique France organization for malaria care and surveillance, the human clinical, epidemiological, and biological data were collected in the French Reference National Center for Malaria (CNRP) database and analyzed in accordance with the public health mission of all French National Reference Centers. The study of the biological samples obtained in the context of medical care was considered as non-interventional research (article L1221-1.1 of the French public health code) and only required the patient's non-opposition during sampling (article L1211-2 of the French public health code).

## DNA extraction and sequencing

Genomic DNA was extracted from each sample using the DNeasy Blood and Tissue Kit (Qiagen, France) according to the manufacturer's recommendations. *P. vivax-like* samples were identified by amplifying with nested PCR the *Plasmodium cytochrome b*, as described in Prugnolle *et al*. [76]. Then, genomic DNA was analyzed on agarose gel (1.8%) in the TAE buffer. Gel-positive amplicons were then sent for sequencing.

For samples identified as *P. vivax*, selective whole-genome amplification (sWGA) was used to enrich submicroscopic DNA levels, as described in Cowell *et al*. [77]. This technique preferentially amplifies *P. vivax* genomes from a set of target DNAs and avoids host DNA contamination. For each sample, DNA amplification was carried out using the strand-displacing phi29 DNA polymerase and *P. vivax*-specific primers that target short (6 to 12 nucleotides) motifs commonly found in the *P. vivax* genome (PvSet1 [77,78]), 30 ng of input DNA was added to a 50-μL reaction mixture containing 3.5 μM of each sWGA primer, 30 U of phi29 DNA polymerase enzyme (New England Biolabs), 1× phi29 buffer (New England Biolabs), 4 mM deoxynucleoside triphosphates (Invitrogen), 1% bovine serum albumin, and sterile water. DNA amplifications were carried out in a thermal cycler with the following program: a ramp down from 35˚ to 30˚C (10 min per degree), 16 hours at 30˚C, 10 min at 65˚C, and hold at 4˚C. For each sample, the products of the two amplifications (one per primer set) were purified with AMPure XP beads (Beckman Coulter) at a 1:1 ratio according to the manufacturer's recommendations and pooled at equimolar concentrations. Samples with the highest concentration of parasite genome were selected after qPCR analysis using a Light Cycler 96 with the following program: 95˚C for 10 minutes; 40 cycles of 95˚C for 15 seconds, 60˚C for 20 seconds and 72˚C for 20 seconds; 95˚C for 10 seconds; and 55˚C for 1 minute. Last, each sWGA library was prepared using the two pooled amplification products and a Nextera XT DNA kit (Illumina) according to the manufacturer's protocol. Each sWGA library was prepared using the two pooled amplification products and a Nextera XT DNA kit (Illumina) according to the

manufacturer's protocol. Samples were then pooled and clustered on a HiSeq 4000 or Novaseq 6000 S4 1 lane PE150 with $2 \times 150$-bp paired-end reads.

## Combining newly sequenced genomic data with those from the literature

The newly sequenced dataset for this study included 74 American *P. vivax* isolates (3 from Brazil, 19 from Ecuador, 22 from French Guiana, 4 from Guyana, 9 from Honduras, and 17 from Venezuela), 63 African isolates (1 from Central Africa, 21 from Comoros, 9 from Djibouti, 1 from Ethiopia, 1 from Gabon, 1 from Mali, 16 from Mauritania, 1 from Mayotte, 8 from Sudan, 2 from Chad, 1 from Uganda, and 1 from an unclear location between Chad and Mali), 40 Middle Eastern isolates (23 from Afghanistan, 4 from Armenia, 11 from Azerbaijan, and 2 from Turkey), 37 South Asian isolates (19 from India, 18 from Pakistan), and 1 Oceanian isolate (PNG).

In addition, genomic data from various sources were added, including those from Daron *et al.* [28], Benavente *et al.* [27], and the MalariaGEN project *P. vivax* Genome Variation [29]. From the large *P. vivax* Genome Variation dataset of the MalariaGEN project (n = 1,895), only samples accessible for which fastq file data were available, passing the quality check defined by Adam *et al.* [29], and that were not related were selected (S1 Fig). The relatedness between haploid genotype pairs within each country was measured by estimating the pairwise fraction of the IBD between strains within populations using the *hmmIBD* program [79], with default parameters for recombination and genotyping error rates, and using the allele frequencies estimated by the program. Isolate pairs that shared >50% of IBD were considered highly related. In each family of related samples, only the strain with the lowest number of missing data was retained. This left 500 samples from the MalariaGEN *P. vivax* Genome Variation project [29]. As the main focus of this study was on Latin America, samples from Malaysia, Myanmar, Thailand, and Cambodia were not retained because we already had more than 20 samples in each country. In total, 295 samples from the MalariaGen *P. vivax* Genome Variation project [31], 499 samples from Daron *et al.* [28], and 125 samples from Benavente *et al.* [27] were selected. The 499 samples from Daron *et al.* [28] included 26 African great apes *P. vivax-like* samples from three African countries (Cameroon, Gabon, and Ivory Coast).

As van Dorp *et al.* [15] inferred a European origin of the American populations, the ancient Spanish sample (Ebro) from this publication was added. Ebro is a composite sample of four samples, from four different slides dating from the 1940s: CA, POS, CM, and lane-8. Samples were downloaded from ENA (study accession PRJEB30878) and the fastq were single ends. The whole dataset from the literature comprised 920 samples that were added to the 214 newly sequenced genomes. Noteworthy, the European samples from Michel *et al.* [16] were not available at the time of the preparation of this dataset.

## *P. vivax* and *P. vivax-like* read mapping and SNP calling

The *P. vivax* and *P. vivax-like* read mapping and SNP calling steps are detailed in S3 Fig. All modern sample sequencing reads were trimmed to remove adapters and preprocessed to eliminate low-quality reads (—quality-cutoff = 30) using *cutadapt* v1.18 [80]. Reads shorter than 50 bp containing "N" were discarded (—minimum-length = 50—max-n = 0). Sequenced reads were aligned to the *P. vivax* reference genome PVP01 [81] with *bwa-mem* v0.7.17 [82].

For Ebro, a different pipeline was used to consider the specific features of ancient DNA. The 3' adapters, reads with low-quality score (—quality-cutoff = 30), and reads shorter than 30 bp were removed with *cuadapt* [80]. Then the cleaned fastq were merged, to create the composite sample as described in van Dorp *et al.* [15]. As the European samples were coinfected by *P. falciparum*, all reads were mapped to the *P. vivax* reference genome PVP01 [81] and to the

*P. falciparum* reference genome Pf3D7 v3 [83] using *bwa-mem* [82]. For reads that mapped to both reference genomes, the edit distance was extracted and reads that aligned equally or better with *P. falciparum* than with *P. vivax* were removed. Then reads with mapping quality <30 and duplicates were removed with *samtools* v1.9 [84] and with *Picard tools* v2.5.0 (broadinstitute.github.io/picard/), respectively. Then, the base quality was rescaled with *Map-Damage 2* [85,86] to account for postmortem damage at the read ends.

For all samples, the *Genome Analysis Toolkit* (*GATK* v3.8.0 [87]) was used to call SNPs in each isolate following the *GATK* best practices. Duplicate reads were marked using MarkDuplicates from the *Picard tools* with default options. Local realignment around indels was performed using the IndelRealigner tool from *GATK*. Variants were called using the HaplotypeCaller module in *GATK* and reads mapped with a "reads minimum mapping quality" of 30 and minimum base quality of >20. During SNP calling, the genotype information was kept for all sites (variants and invariant sites, option -ERC) to retain the information carried by the SNPs fixed for the reference allele. Then only the nuclear genome was kept, filtered out with *BCFtools* v1.10.2 [84,88], and all individual VCF files were merged with *GATK*. Organelle genomes (mitochondria and apicoplast) were not included because they are haploid markers, without recombination and with only maternal heritability [89–91].

## Data filtering

Combining data from the literature (n = 919) with the 214 newly sequenced samples resulted in a total of 1,133 samples for the unfiltered dataset: 1,106 modern human *P. vivax* samples, 26 modern African great apes *P. vivax-like* samples, and one ancient human *P. vivax* sample. All filtration steps are detailed in S1 Fig.

For all modern samples, samples with >50% missing data were removed. As several strains can infect the same host, the within-host infection complexity was assessed with the $F_{WS}$ metric [92], calculated with *vcfdo* (github.com/IDEELResearch/vcfdo; last accessed July 2022). Samples with mono-clonal infections, *i.e.* $F_{WS}$ >0.95 were kept. Highly related samples and clones could have generated spurious signals of population structure, biased estimators of population genetic variation, and violated the assumptions of the model-based population genetic approaches used in this study [93]. The relatedness between haploid genotype pairs was measured by estimating the pairwise fraction of the genome IBD as explained above. Within each country, isolate pairs that shared >50% of IBD were considered highly related. In each family of related samples, only the strain with the lowest number of missing data was retained. The final dataset included 619 modern *P. vivax* individuals from 34 countries, two *P. vivax-like* from Cameroon, infecting Nigeria-Cameroon chimpanzees (*Pan troglodytes ellioti*), and Ebro, an ancient DNA sample from Spain.

Two dataset formats were used in function of the analysis: (*i*) a VCF containing all the samples, and (*ii*) bam files of the samples for the *ANGSD* v0.940 [30] or *ANGSD* suite software (S1 Fig). For both formats, the analysis focused only on the core genome regions as defined by Daron *et al.* [8].

For the VCF, only bi-allelic SNPs with ≤20% missing data and a quality score ≥30 were kept. The minimum and maximum coverage were 10 and 106. Moreover, singletons were removed to minimize sequencing errors. Finally, 950,034 SNPs with a mean density of 41.14 SNPs per kilobase were retained.

For the *ANGSD* suite software, all sites with a base quality ≥20, a maximum total depth of 106 X, a minimum total depth of 5 X, and present in at least five individuals were kept. For analyses that did not require invariant sites, only bi-allelic SNPs with a *p-value* = 1e-6 were

retained. In total 20,844,131 sites from the core genome were considered, including 1,437,273 variants (69.59 SNPs/kb).

## Population structure

The PCA and the ancestry plots were computed with *ANGSD* and *PCAngsd* v0.98, after selecting only biallelic SNPs present in the core region of the *P. vivax* genomes and excluding SNPs with a minor allele frequency (MAF) ≤5%. The variants were LD-pruned to obtain a set of unlinked variants using *ngsld* v1.1.1 [94], with a threshold of $r^2 = 0.5$ with windows of 0.5kb. In total, 105,527 SNPs were used for 620 individuals (*P. vivax-like* samples were removed). For the ancestry plots, obtained from *PCAngsd*, the k value (number of clusters) was from 2 to 16. Then, *pong* v1.5 [95] was used to analyze the *PCAngsd* outputs and compute ancestry proportions.

The ML tree was obtained with *IQ-TREE* v2.0.3 [34] using the best-fitted model determined by *ModelFinder* [34]. As the dataset comprised 950,034 SNPs of the core genome and no constant site, the ascertainment bias correction was added to the tested models. The best inferred model was a general time reversible (GTR) model of nucleotide evolution that integrated unequal rates and unequal base frequency. The node reliability was assessed with Ultrafast Bootstrap Approximation [96] and the SH-aLRT test [97].

## Relationships between populations

Population networks were estimated using two different approaches: *TreeMix* v1.13 [39] and *AdmixtureBayes* (github.com/avaughn271/AdmixtureBayes; last accessed June 2023) [40]. Unlike with the "classic" ML phylogenetic tree at the level of individual strain genomes (see above), these two methods use allele frequency (co-)variation within and among populations to derive the fraction of shared versus private genetic ancestry (or genetic drift) and to infer the population branching order, while taking into account population genetic processes, such as genetic drift, historical migration, and admixture events between populations [98,99].

To be able to compare results, the same dataset was used for both approaches: the VCF dataset was restricted to SNPs without missing data in Ebro (17,696 variants) and to subsampled modern populations as follows. Only populations with the largest sample sizes for each genetic cluster (according to PCA and ancestry plots) were kept, as well as all the American populations. The SNP data set was LD-pruned with With *PLINK* v2 [100], a threshold of $r^2 = 0.5$, with sliding windows of 50 SNPs and a step of 10 SNPs, and MAF filtering with a threshold of 5% (9,022 SNPs remaining). In some populations, there were still positions with missing data, and those positions were removed because *TreeMix* do not handle missing data. This filtering resulted in 7,109 SNPs for 18 ingroup populations (from n = 1 to n = 80) and one outgroup population (*P. vivax-like*).

For the *TreeMix* analyses, the number of migration events ($m$) that best fitted the data were estimated by running *TreeMix* 15 times for each $m$ value, with $m$ ranging from 1 to 10. The optimal $m$ value was estimated using the *OptM* R package [101]. Then, a consensus tree with bootstrap node support was obtained by running *TreeMix* 100 times using the optimal $m$ value. Results were post-processed using the *BITE* R package [102].

For the *AdmixtureBayes* analyses, three independent runs were performed, each including 40 Markov Chain Models and 250,000 steps, to identify convergence. Convergence was checked with a Gelman-Rubin Plot, as recommended by Nielsen *et al.* [40]. This analysis found that two chains had converged (S7 Fig). The results of these two chains were analyzed after a burn-in of 50% and a thinning step of 10, using the default parameters. The common

network among the three best networks in each chain that converged was retained, based on posterior probabilities.

From the dataset generated with *PLINK* for *TreeMix* and *AdmixtureBayes*, the $f_3$-statistics were calculated using *ADMIXTOOLS 2* v2.0.0 [103].

## Demographic and invasion history of *P. vivax* in the Americas

Tajima's D [43] and nucleotide diversity (π) were measured for each population with ≥10 individuals to avoid biases [104]. The sample size was standardized at 10 (*i.e.*, 10 randomly chosen isolates for each population) to obtain values that could be compared. Tajima's D and nucleotide diversity (π) values were estimated using *ANGSD* [30], with a window of 500 bp and a step of 500 bp in the core genome, and only windows with ≥50 sites were kept.

The coalescent-based approach of *Relate* [44] and the core region of the genome of all chromosomes were used to infer historical changes in $N_e$ and to estimate the divergence time between modern populations. *Collate* was used to calculate CCR between Ebro and modern populations, because this program was designed to handle ancient DNA samples with low coverage [47]. For both analyses, a mutation rate of 6.43 x $10^{-9}$ mutations/site/generation [28] and a generation time of 5.5 generations/years [28] were used. Alleles were polarized using the two *P. vivax-like* samples as an outgroup.

A genetic map was required to carry out these analyses. This was generated using *LDhat* [105] and considering the Thailand population, one of our most diverse populations. First, individuals with ≥5% missing data were removed, leaving 58 samples. The function *pairwise* was used to estimate a first approximation of Watterson's Theta and then, the likelihood table was estimated with these parameters: *-n 58 -rhomax 100 -n_pts 101 -theta 0.00011 -split 8*. The recombination map was then obtained by using the function *interval* (*-its 1100000 -samp 100 -bpen 5*) followed by *stat* with a burn-in of 10,000 samples. The recombination map ($4N_e$r/pb) was converted into a genetic map (cM/Mb) with a $N_e$ estimate of 6,000 (inferred with *Stairwayplot 2* on chromosome 14).

## Demographic scenarios testing of *P. vivax* invasion of the Americas using DIYABC-RF

Different possible scenarios of population divergence and admixture were evaluated using *DIYABC Random Forest* (RF) [49] via the *diyabcGUI* v1.2.1 R package.

As Ebro is a composite sample, it was made haploid by creating two haploid samples, splitting the "heterozygous" sites between the two samples. To minimize the effect of missing data, the sites genotyped in at least one individual in each population and at least in 40% of individuals in all populations combined were kept. Moreover, individuals with ≥ 50% missing data were removed. Thus, 45 haploid individuals for three populations and 3,139 SNPs were retained.

Twelve scenarios were tested: (1) American populations as a result of West African and European admixture; (2) uniquely European origin; (3) uniquely West African origin; (4) Ebro as a migrant from Latin America with American populations descending from West African ancestors; (5) Ebro resulting from the admixture between American populations and an unsampled West African "ghost" population; the American populations would have split from an ancestral population; (6) American populations stemming from the admixture of European and unsampled populations; (7) American populations from Europe with a secondary West African contribution; (8) American populations from West Africa with a secondary European contribution; (9) American populations from Africa with two introduction waves; (10) American populations originating from Europe, with Ebro resulting from an American-European

ghost population admixture; (11) American populations resulting from two successive admixture events, the first and oldest one between the European populations and an unsampled ghost population, followed by a more recent wave of introduction from West Africa; and (12) Ebro resulting from the American-ghost population admixture, with American populations splitting from West Africa.

The scenario parameters were considered as random variables drawn from prior uniform distributions (S3 Table). *DIYABC-RF* was used to simulate 20,000 genetic datasets per scenario with the same properties as the observed data set (number of loci and proportion of missing data). Simulated and observed datasets were summarized using the whole set of summary statistics proposed by *DIYABC-RF* for SNP markers to describe the genetic variation of each population (e.g., proportion of monomorphic loci, heterozygosity), pair of populations (e.g., $F_{ST}$ and Nei's distances), or trio of populations (e.g., $f_3$-statistics, coefficient of admixture). The total number of summary statistics was 50.

Then with these 20,000 simulated data sets per scenario, the *RF* classification procedure was used to compare the likelihood of the competing scenarios. *RF* is a supervised machine-learning algorithm that uses hundreds of bootstrapped decision trees to perform classifications, using the summary statistics as a set of predictor variables. Then a classification forest of 1500 trees was grown.

Some simulations were excluded from the decision tree building during each bootstrap (*i.e.*, out-of-bag simulations). These were used to calculate the prior error rate and the type II error rate. Additionally, the confusion matrix was computed to provide a more global assessment of the RF procedure performance [106] (S10 Fig). The result convergence was evaluated over ten independent RF runs, as recommended by Fraimout *et al.* [107]. To determine the compatibility of the formulated scenarios and associated priors with the observed dataset, we plotted the summary statistics for both on the first two principal axes of a linear discriminant analysis and of a PCA (S10 Fig), following the recommendation by Pudlo *et al.* [50]. The location of the summary statistics from the observed data within the clouds formed by those from the simulated data was visually checked to ensure that the simulations under the tested scenarios are realistic enough to reproduce the observed dataset (S10 Fig).

The RF computation provides a classification vote for each scenario (*i.e.*, the number of times a model is selected from the decision trees). The scenario with the highest classification vote was selected as the most likely scenario.

Then, the posterior distribution values for all parameters for the best model identified were estimated using the derivation of a new RF for each component of interest of the parameter vector [108], with classification forests of 1500 decision trees, and based on a training set of 100,000 simulations. Estimated parameters included the effective size ($N_e$) for each population, split times among populations, and admixture event timing and rates. Estimates for timing parameters were converted from generations to years assuming a generation time of 5.5 generations per year [28].

## Supporting information

**S1 Fig. Filtering and dataset creation.** Each box in the diagram represents a specific filtering step and also indicates the number of remaining samples or SNPs. Key filtering options also are indicated. The final steps (dotted line boxes) show the analyses performed using the different datasets.
(TIF)

**S2 Fig. Within-sample infection complexity ($F_{WS}$ index) and inbreeding (identity by descent) in *P. vivax* and *P. vivax-like* populations.** The $F_{WS}$ index is a proxy of the diversity

within individual infections, from 0 (high diversity) to 1 (no diversity). $F_{WS}$ values >0.95 (indicated by a dotted line) usually indicate monoclonal infections. Identity by descent (IBD) indicates the percentage of the genome resulting from inbreeding among pairs of individuals of the same population. A pairwise IBD >0.5 (dotted line) resulted in the exclusion of one of the two individuals in the strain pair. CAF = Central Africa, WAF = West Africa. PNG = Papua New Guinea.

(TIF)

**S3 Fig. *P. vivax* and *P. vivax-like* read mapping and SNP calling steps.** The steps in blue are specific to the ancient DNA Ebro sample, while the steps in red are specific to modern samples. Steps in purple are common to both modern and ancient samples. The gray steps highlight the datasets in the two formats used in this study: bam and VCF.

(TIF)

**S4 Fig. Principal component analysis (PCA) for 619 modern *P. vivax* strains and the Ebro ancient DNA sample from Spain based on the genotype likelihood of 105,527 unlinked SNPs. (a)** Percentage of variance explained by the first 15 principal components (PC). The optimal number of PCs is 5, as determined by the elbow (broken-stick) method. **(b)** PCA plots for PC 1 to 5. PNG: Papua New Guinea.

(TIF)

**S5 Fig. Genetic ancestry of *P. vivax* populations worldwide estimated with *PCAngsd* (K = 2 to K = 16).** The number (K) of clusters tested is specified on the left. According to Meisner and Albrechtsen [31], the best K is determined by 1+ D (the optimal number of principal components). As presented in S4 Fig, D would be = 5 (determined by the elbow (broken-stick) method), thus K = 6. PNG: Papua New Guinea.

(TIF)

**S6 Fig. Identification of the optimal number of migration edges (*m*) for 18 *P. vivax* populations and consensus topology for m = 5 with *TreeMix*. (a)** Changes in the mean likelihood score (± SD) and the mean total fraction of the genetic variance explained (± SD) in function of the number of migration edges in the *TreeMix* analysis. (b) The second-order rate of change in likelihood (Δ*m*) across migration edges (*m*) values. The *OptM* R package [101] and the Evanno method were used for panels A and B, using 15 replicates for each migration edge (*m*) value, from 0 to 10. The inflection points were observed at *m* = 2 and *m* = 5. (c) *TreeMix* tree of a subset of 18 *P. vivax* populations with five migration edges (arrows), rooted with *P. vivax-like* indicated with the asterisk. The scale bar shows ten times the mean standard error (s.e.). The migration weight is indicated with a color scale, from yellow (0%) to red (50%). (d) Visualization of the residuals from the fit of the model to the data for the trees with *m* = 2 and *m* = 5. The color scales indicate the residuals in standard error units, red for positive values and blue for negative values. PNG: Papua New Guinea.

(TIF)

**S7 Fig. Identification of the optimal number of admixture events for 18 *P. vivax* populations and consensus topology with *AdmixtureBayes*. (a)** Trace plots of the posterior probability, total branch length, and number of admixture events for each chain. **(b)** The plot of the Gelman-Rubin convergence diagnostics on chains 1 and 2 for three summary statistics after a burn-in fraction of 50%. The rapid convergence to 1 indicates that this is a sufficient burn-in period. (c) Consensus tree generated by combining nodes with a posterior probability >25% of appearing in the admixture graph. Square nodes represent admixture events. The percentages in the nodes are the posterior probability that the true graph has a node with the same

descendants. PNG: Papua New Guinea.
(TIF)

**S8 Fig. $f_3$-statistics for American populations with possible source populations from other American populations and other *P. vivax* populations of the world (Africa, Asia, and Europe).** The dotted line marks the zero. All values are not significantly negative. PNG: Papua New Guinea.
(TIF)

**S9 Fig. Genetic diversity of *P. vivax* populations.** Distributions of the nucleotide diversity ($\pi$) and Tajima's D values for populations with >20 samples. The same superscript letters indicate no significant difference between the distributions (Wilcoxon test and Bonferroni correction). If there is no superscript, the distribution is significantly different from all other distributions with a *p-value* < 0.003.
(TIF)

**S10 Fig. ABC-RF performance to select the best model. (a)** Principal Component Analysis (PCA) based on the summary statistics generated by the simulated datasets from the training set (a color indicates each simulated model) and the observed dataset (indicated with a black diamond). **(b)** Projection of the reference table on the first two Linear Discriminant Analysis (LDA) axes. Colors correspond to the model indices. A black diamond indicates the location of the observed dataset. **(c)** ABC-RF out-of-bag confusion matrix. The diagonal represents the proportion of simulated datasets correctly classified for each demographic scenario. Each column corresponds to the scenario under which simulations were generated and each row is the best-supported scenario selected by the ABC-RF classifier. The values represent the mean number of votes over 10 replicates, also represented by the color gradient (the darker the color, the more votes). The minimum and maximum number of votes is indicated between brackets.
(TIF)

**S1 Table. Samples metadata and listing.** The NCBI SSR-ID, bioproject, biosample, and source are indicated for each sample. When available, the latitude and longitude are specified. NA, information not available. The QC column indicates whether samples have successfully passed the quality control (QC) and are included in the final dataset for analyses. For samples that did not pass the QC, the reasons are outlined in the "Reasons_QC" column as follows: "Missing data" for samples with >50% missing data, "Low $F_{ws}$" for samples removed due to multi-clonal infections, and "High IBD" for samples that are related to those kept in the analysis dataset. For additional details, please refer to the Materials and Methods section and S2 Fig.
(XLSX)

**S2 Table. Metadata for the analysis dataset.** For each sample, the number of reads and the mean coverage are indicated. Each sample metadata included the percentage of the genome covered by at least 1X (% > 1X), 5X (% > 5X), and 10X (% > 10X) sequencing depth. When available, the latitude and longitude are specified. NA: information not available.
(XLSX)

**S3 Table. Distribution and conditions of the DIYABC-RF scenario parameters.** To see the correspondence between the parameters and the scenarios, refer to Fig 5a. All distributions are uniform.
(PDF)

**S4 Table. Results of the model selection using DIYABC-RF, with French Guiana as the representative population of Latin America.**
(PDF)

## Acknowledgments

The authors acknowledge the ISO 9001-certified IRD i-Trop HPC (member of the South Green Platform) at IRD Montpellier for providing HPC resources that contributed to the research results reported within this paper. URL: https://bioinfo.ird.fr/- http://www.southgreen.fr. The bioinformatics analyses were also performed on the Core Cluster of the Institut Français de Bioinformatique (IFB) (ANR-11-INBS-0013).

The authors also thank the members of the French National Reference Centre for Imported Malaria Study Group for providing *P. vivax* isolates. We would like to thank Clarice Moulin for his help in the laboratory work on the newly sequenced samples and to Josquin Daron for providing the scripts for mapping and calling the samples. We also thank Leo Speidel for his guidance on *Relate* and *Colate* analyses and Arnaud Estoup for his support with DIYABC-RF analyses. Finally, we thank Elisabetta Andermarcher for proofreading the manuscript and supplementary material.

## Author Contributions

**Conceptualization:** Margaux J. M. Lefebvre, Franck Prugnolle, Michael C. Fontaine, Virginie Rougeron.

**Data curation:** Fanny Degrugillier, Céline Arnathau.

**Formal analysis:** Margaux J. M. Lefebvre.

**Funding acquisition:** Fabian E. Sáenz, Franck Prugnolle, Michael C. Fontaine, Virginie Rougeron.

**Investigation:** Gustavo A. Fontecha, Oscar Noya, Sandrine Houzé, Carlo Severini, Bruno Pradines, Antoine Berry, Jean-François Trape, Fabian E. Sáenz.

**Project administration:** Virginie Rougeron.

**Resources:** Fanny Degrugillier, Céline Arnathau, Virginie Rougeron.

**Supervision:** Franck Prugnolle, Michael C. Fontaine, Virginie Rougeron.

**Visualization:** Margaux J. M. Lefebvre.

**Writing – original draft:** Margaux J. M. Lefebvre, Franck Prugnolle, Michael C. Fontaine, Virginie Rougeron.

**Writing – review & editing:** Margaux J. M. Lefebvre, Franck Prugnolle, Michael C. Fontaine, Virginie Rougeron.

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
