## [Decision Letter · Decision Letter 0]

22 Aug 2024

Dear Ms Lefebvre,

Thank you very much for submitting your manuscript "Genomic exploration of the complex journey of Plasmodium vivax in Latin America" for consideration at PLOS Pathogens. As with all papers reviewed by the journal, your manuscript was reviewed by members of the editorial board and by several independent reviewers. In light of the reviews (below this email), we would like to invite the resubmission of a significantly-revised version that takes into account the reviewers' comments.

We cannot make any decision about publication until we have seen the revised manuscript and your response to the reviewers' comments. Your revised manuscript is also likely to be sent to reviewers for further evaluation.

Sincerely,

Kenneth D Vernick

Academic Editor

PLOS Pathogens

Francis Jiggins

Section Editor

PLOS Pathogens

Michael Malim

Editor-in-Chief

PLOS Pathogens

orcid.org/0000-0002-7699-2064

Reviewer's Responses to Questions

**Part I - Summary**

Reviewer #1: This manuscript by Lefebvre and colleagues describes the analysis of Plasmodium vivax malaria parasite genomic data to understand the manner in which it was introduced to the Americas. The authors analyzed data from 513 previously sequenced P. vivax samples, and contributed high quality sequencing data from 107 newly sequenced samples, many of which come from previously undersampled (or unsampled) populations, such as Mauritania, Cameroon, Comoros, Sudan, and Central African Republic. With these data, the authors suggest that the current reigning hypothesis (that American P. vivax populations represent an extirpated European clade brought across the Atlantic in the colonial period) is too simple. They propose that the American P. vivax populations exhibit ancestry from W. Africa, as well potentially from other regions such as South Asia.

The authors make the most of sometimes thin data (eg fewer than 10 samples from many countries). The dataset they have assembled reflects similar global population structure as observed in the Pv4 (2002) publication, as well as predecessor publications of comparable global Pv genome datasets. The authors analyse the data more deeply with regard to population structure and recent demography than the Pv4 manuscript, however, making these results of interest to a wide audience. The manuscript is very clearly written.

Reviewer #2: The substantial number of newly sequenced Plasmodium vivax isolates in this dataset is a significant contribution to the malaria research community. However, considering a broader context for analyzing this dataset could potentially enhance the contribution of this work to science. Nevertheless, this study, while incremental, contributes to the field with its data, which are definitively of value.

The rejection of a “naïve” single origin followed by a severe bottleneck model seems compatible with the data. A potential European origin of the populations found in the Americas was previously discussed (Culleton et al. 2011, Taylor et al. 2013). Adding the data from the article by van Dorp that studies a particular isolate using ancient DNA (cited by the authors) was necessary because the original paper was somehow limited in its analyses. However, it is also essential to include the data and work by Michel et al. 2024 (doi.org/10.1038/s41586-024-07546-2). This latest study was excluded from their discussion, and it reached interesting conclusions as it highlighted more the European postcolonial contacts of a diverse population. Perhaps this work was finished before the Michel et al. 2024 paper became available.

Unfortunately, as presented, the authors seem to reproduce the main results of Michel et al. 2024 with different methods but less data from Europe. The strong geographic population structure exhibited by P. vivax, not only in The Americas but worldwide, is definitively interesting but, again, has been reported by others. The data, however, is of great importance to the malaria research community, so I encourage the authors to address the diversity of P. vivax with a broader scope than just focusing on The Americas.

Reviewer #3: The authors employ complimentary approaches – population genetics (structure, diversity, ancestry) and coalescent based scenario testing to infer the evolutionary history of Plasmodium vivax in Latin America with particular emphasis on when and from where it most likely migrated to the region. A number of other studies using mitochondrial or microsatellite markers have addressed this question, variously implicating Melanesia, Africa, Europe, and South-Asia. The current study employs an expanded number of P. vivax genomes (620), including isolates from West Africa, and one ancient DNA sample from Spain. The authors conclude that the current P. vivax populations in Latin America primarily originated from an extinct European lineage, as well as an unsampled population, which they conclude to be of likely West African origin.

In general the population genetics analyses are thorough and provide a strong basis for the conclusion that Latin American P. vivax forms a distinct group without strong evidence of admixture with other regional populations, and that the Ebro isolate most closely resembles the Latin American population. Divergence time estimates between Latin American and other populations are most recent for the Ebro sample and West Africa. Twelve ancestry scenarios featuring the Ebro isolate, Mauritanian, and Colombian populations were tested using a simulation-based approximate Bayesian computation (ABC) statistical framework that relies on the Random Forest tree classifier (ABC-RF).

Some results have been over-interpreted (see specific comments below). Additionally, the organizational structure of the manuscript could be improved. Arguments and conclusions are repeated in great detail in several sections (introduction, Results, Discussion) of the manuscript. In particular, the Discussion section could be condensed and several subheadings removed.

**Part II – Major Issues: Key Experiments Required for Acceptance**

Reviewer #1: My central contention with the paper is that given the limitations of sampling, and the weakness of the signals they are attempting to measure, the authors cannot refute the most parsimonious and simple hypothesis that P. vivax in the Americas is derived almost exclusively from the extirpated European population, despite suggestive language to the contrary in the title and throughout. As noted by the authors around line 263, the EBRO sample from Spain dating from the 1940s shows a similarly complex potential ancestry as the Latin American samples by the PCAngsd analysis in Figure 2c. If the authors cannot address the possibility that the European source population (if typified by EBRO) was already admixed, it seems unwarranted to speculate that there was a ‘complex journey’ of Pv to Latin America. All the parasites could have come from Europe. Period. This possibility is acknowledged again by the authors in the context of the f3 analysis on line 308, but not given adequate attention as a parsimonious scenario in the discussion, which focuses on insights from ABC-RF and other fancy approaches. Further, the admixture signal from W. Africa in Figure 2c appears much stronger at K=6 than K=9. It is unclear to what degree the central novel result of this paper is driven by the fragile choice of a K value.

The authors make the valid point that the EBRO data are a composite from multiple specimens, and are of poor quality. However, the recent publication of the ancient DNA analysis by Michel and colleagues in Nature last week fortifies the signal from EBRO, and unfortunately renders much of the text in the Introduction and Discussion sections of this manuscript in need of updating. The Michel manuscript finds strong affinity between EBRO, ancient European Pvivax genomes, contemporary P.vivax genomes from the Americas, as well as a Pvivax genome from Peru that dates back 500 years. The Michel manuscript worked with previously published genomes from Mauritania, Madagascar, and Eritrea (as well as Ethiopia) but found no evidence of admixture in European/American Pv genomes. This disparity in outcomes should be addressed by the authors. Is it a product of fewer W. African genomes in the Michel analysis? Or fewer genomes from French Guiana, Guyana, and Venezuela? Or ?

Specific Points:

As noted above, the introduction and discussion (if not the dataset and analysis itself) need significant revision in light of the Michel paper for this paper to maintain relevance. For example, the first sentence of the Conclusion section (that Pv was introduced to the Americas in the 19th century) is now strongly unsupported by the data from the 500 year old Chachapoyas (Peru) sample in the Michel paper. The analyses underlying this conclusion, and their underlying assumptions, should be revisited.

The title and language used to describe the origin of the American Pvivax population should be toned down further, unless the evidence for a complex journey can be strengthened.

Line 611: under what plausible historical human scenario could the European, West African, and ‘unsampled’ P.vivax populations have all diverged at the same time (400 years ago)? Could this imply that W. African Pv populations were also introduced from Europe during the colonial era? The ABC-RF analysis feels over-interpreted given the light depth of sampling of many regions and the departures of plasmodium from Wright-Fisher expectations (including generations that occur without outcrossing when monoclonal infections are transmitted, which should be mentioned)

Reviewer #2: Overall, the article offers limited new insights by focusing just on the origin of P. vivax in the Americas. The sampling from the Americas, Africa, and Asia corroborates previous findings of strong population structure. However, the paper does not include data from potential African areas where vivax could have been imported into the Americas nor populations that could have contributed to the now-extinct European populations limiting their capacity to explore complex colonization scenarios. Thus, the statement that this work “represents nearly all potential source populations worldwide” is inaccurate and the limitation of the sampling should be acknowledged.

Considering the impact of PloS Pathogens, the data from Michel et al. 2024 must be included in any analysis. I see no convincing support for a West African contribution, so I suggest the authors elaborate more. The conclusion seems to have limited backing from data and more from the assumptions and scenarios tested in the model. For example, they could put the magnifier in other hypothetical colonizations, such as including populations like Ethiopia or India, and see how the models behave. Also, it is important to include others from The Americas, like Brazil and Guyana, not just Colombia (the analysis with DIYABC-RF is interesting but also the methodology from Speidel et al. allows us to address migration, perhaps comparing results is of interest). If they find the colonization from West Africa, then the result will be more robust; my concern is that they are observing connections in the universe of samples they choose to model, which may not capture reality.

“Europe” could have been particularly isolated from other areas; but there was commerce, so there could have been gene flow and population structures in pre-colonial times. Most of those contacts with Europe were excluded from the colonization pathway (the focus on the Americas perhaps biased the big picture). Considering that Europe has such importance, the data from Michael et al. 2024 is crucial.

What is clear, from the strong population structure reported here and elsewhere, is that the separation of The Americas was expected and comparable with the separation of Southeast Asia from Africa, the Indian subcontinent, etc. The limited data from Bangladesh seems separated from India, but India is a subcontinent, so “India” as a geographic location is misleading. The same applies to Brazil and Colombia, which have complex geographic landscapes and highly structured populations. Some areas of the Amazonia in Brazil could be easily connected with Peru, Colombia, and Ecuador. In contrast, others are more clearly associated with Venezuela and Guyana because of gold mining in the Guyana Shield. The Pacific Coast of Colombia is expected to relate to Central America. Perhaps focusing on other results can enrich the impact of the paper.

All extant vivax populations appear to be relatively modern with these analyses; maybe the authors could elaborate. The rationale for the mutation rate and number of generations per year need to be clarified since it determines the time estimates, perhaps exploring other scenarios is important. The method developed by Speidel et al. allows for testing scenarios with different mutation rate estimates, etc. However, I'm not sure how the hypnozoite affects the modeling framework implemented in terms of overlapping generations. Perhaps some simulated data could be used to test its impact on the estimates yielded by the methods.

Reviewer #3: (No Response)

**Part III – Minor Issues: Editorial and Data Presentation Modifications**

Reviewer #1: Small points

Extra use of the word ‘countries’ in line 126

Reviewer #2: The data is important but also is being self critical of its limitations. Data from the middle East is lacking still so encouraging the readers toward exploring the sampling gaps could be of interest. It is also an overstatement to say that all potential sources are considered since the Gordian knot seems to be Europe.

Please provide the bioproject number in the maintext.

Reviewer #3: Line 65. The authors state that “knowledge of when and from where P. vivax colonized Latin America is “necessary to develop efficient strategies for malaria control and elimination and for monitoring its spread in new regions” yet the rationale for this statement is lacking and it seems to be a bit of a stretch.

Line160-169. More appropriate in the Methods section.

Figure 2a.

•A 3-axis PCA plot without animation (rotation) is difficult to interpret visually.

•Labels – Cameroon and Central Africa are grouped with Mauritania under the heading West Africa and yet they are central African.

•Not clear why only 105,527 unlinked SNPs used rather than 950,034

Figure 2d. If K=6 is the best K-value based on the broken-stick eigenvalues plus one method, why does 2d illustrate K=9?

Line 245. Results from PC4 are discussed but this PC only explains 2.28% of the variation.

Line 303-311. It is not clear what the admixture fs-statistics add to the analysis as the results are inconclusive.

Line 394. “These divergence time estimates are consistent with the population branching order from the previous analyses (Fig. 2b and 3)”. Figure 2b does not provide strong basal branch support and therefore cannot be said to be consistent with divergence time estimates.

The authors acknowledge that the population genetics analyses provide only weak support for a West African contribution to genetic ancestry of P. vivax populations in Latin America (Fig 2C).

Colonization history scenarios. Why was the Colombian population chosen? Do the results differ if different Latin American populations are used.

Line 453. What were the mean posterior probabilities of scenarios 4 and 8?

Fig 3. Other than highlighting the distinctness of the Latin American populations, what does this analysis add?

Fig 5. Labels are not defined. What is Nbot?

Line 467. Given the large 90% confidence interval associated with estimated divergence it seems reporting specific years for the divergence times is a bit of an over-interpretation.

Discussion

This section would benefit from significant editing. As it is, many of the arguments and results reported in previous sections (Introduction, Results) are repeated here. A clear synthesis of the salient results would be preferable.

PLOS authors have the option to publish the peer review history of their article (what does this mean?). If published, this will include your full peer review and any attached files.

Reviewer #1: No

Reviewer #2: No

Reviewer #3: No
---

## [Editor Report · Decision Letter 1]

5 Dec 2024

Dear Ms Lefebvre,

We are pleased to inform you that your manuscript 'Genomic exploration of the journey of Plasmodium vivax in Latin America' has been provisionally accepted for publication in PLOS Pathogens.

Best regards,

Kenneth D Vernick

Academic Editor

PLOS Pathogens

Francis Jiggins

Section Editor

PLOS Pathogens

Sumita Bhaduri-McIntosh

Editor-in-Chief

PLOS Pathogens

orcid.org/0000-0003-2946-9497

Michael Malim

Editor-in-Chief

PLOS Pathogens

orcid.org/0000-0002-7699-2064
---

## [Editor Report · Acceptance letter]

5 Jan 2025

Dear Ms Lefebvre,

We are delighted to inform you that your manuscript, "Genomic exploration of the journey of Plasmodium vivax in Latin America," has been formally accepted for publication in PLOS Pathogens.

Best regards,

Sumita Bhaduri-McIntosh

Editor-in-Chief

PLOS Pathogens

orcid.org/0000-0003-2946-9497

Michael Malim

Editor-in-Chief

PLOS Pathogens

orcid.org/0000-0002-7699-2064